# Ssl2/TFIIH function in transcription start site scanning by RNA polymerase II in *Saccharomyces cerevisiae*

**Tingting Zhao[1†], Irina O Vvedenskaya[2], William KM Lai[3], Shrabani Basu[1], B Franklin Pugh[3], Bryce E Nickels[2], Craig D Kaplan[1]***

[1]Department of Biological Sciences, University of Pittsburgh, Pittsburgh, United States; [2]Department of Genetics and Waksman Institute, Rutgers University, Piscataway, United States; [3]Department of Molecular Biology and Genetics, Cornell University, Ithaca, United States

**\*For correspondence:**
craig.kaplan@pitt.edu

**Present address:** [†]Genomics and Bioinformatics Hub, Brigham and Women's Hospital, Harvard Medical School, Boston, United States

**ABSTRACT** In *Saccharomyces cerevisiae*, RNA polymerase II (Pol II) selects transcription start sites (TSSs) by a unidirectional scanning process. During scanning, a preinitiation complex (PIC) assembled at an upstream core promoter initiates at select positions within a window ~40–120 bp downstream. Several lines of evidence indicate that Ssl2, the yeast homolog of XPB and an essential and conserved subunit of the general transcription factor (GTF) TFIIH, drives scanning through its DNA-dependent ATPase activity, therefore potentially controlling both scanning rate and scanning extent (processivity). To address questions of how Ssl2 functions in promoter scanning and interacts with other initiation activities, we leveraged distinct initiation-sensitive reporters to identify novel *ssl2* alleles. These *ssl2* alleles, many of which alter residues conserved from yeast to human, confer either upstream or downstream TSS shifts at the model promoter *ADH1* and genome-wide. Specifically, tested *ssl2* alleles alter TSS selection by increasing or narrowing the distribution of TSSs used at individual promoters. Genetic interactions of *ssl2* alleles with other initiation factors are consistent with *ssl2* allele classes functioning through increasing or decreasing scanning processivity but not necessarily scanning rate. These alleles underpin a residue interaction network that likely modulates Ssl2 activity and TFIIH function in promoter scanning. We propose that the outcome of promoter scanning is determined by two functional networks, the first being Pol II activity and factors that modulate it to determine initiation efficiency within a scanning window, and the second being Ssl2/TFIIH and factors that modulate scanning processivity to determine the width of the scanning widow.

## Introduction

Transcription of eukaryotic protein-coding genes is carried out by RNA polymerase II (Pol II) in three sequential steps: initiation, elongation, and termination (*Roeder, 2019*). Accurate initiation requires minimally the assistance of five general transcription factors (GTFs) TFIIB, TFIID, TFIIE, TFIIF, and TFIIH, which together with Pol II, comprise the basal transcription machinery. At the beginning of transcription, this machinery assembles at a defined DNA region for each transcript called a promoter, melts the double-stranded DNA yielding a region of unwound DNA forming a 'transcription bubble'. Within this bubble a position or positions will be identified to serve as transcription start sites (TSSs). Initial promoter melting appears to occur stereotypically ~20–25 nt downstream of promoter elements such as the TATA box across eukaryotes, though most promoters lack a TATA box or other strong sequence signature. During initiation, the process of TSS selection determines the identity and distribution of transcript isoforms that differ by their 5' ends. Differences in 5' UTR can alter transcript properties

**eLife digest** In eukaryotic organisms such as yeast, the process of converting genes into proteins begins with the transcription of DNA sequences into mRNA molecules. An enzyme called RNA Polymerase II (Pol II) is responsible for creating new strands of mRNA, but a variety of other so called transcription factors is also needed to kickstart the transcription process. These transcription factors are delivered to genes, where they attach to specific sequences, or promoters, which sit at the beginning of each gene.

Once these transcription factors are in place, the double stranded DNA is unzipped to provide access to the DNA that will serve as the template for transcription. In budding yeast, Pol II and another specific transcription factor, known as TFIIH, work together to scan these promoter sequences to find the appropriate start sites of mRNA synthesis. However, several aspects of this process, such as how TFIIH works in promoter scanning, how far its scanning functions can extend, and how its activity is controlled, are currently poorly understood.

Zhao et al. have investigated these questions in budding yeast. Using a range of genetic and genomic techniques, Zhao et al. found that certain sections of TFIIH were involved in choosing specific transcription start sites of mRNA synthesis during promoter scanning. These sections were identical in different eukaryotic organisms from yeast to humans, suggesting that these regions may be important for tuning or controlling the activity of TFIIH. Moreover, in yeast, the activity of TFIIH determines how far the scanning unit was able to move along the promoter DNA.

Finally, Zhao et al. found that the initiation by promoter scanning was regulated by two distinct networks. The first network controlled how well mRNA synthesis could be initiated at individual transcription start sites; and the second network – driven by TFIIH – controlled which promoter sequences could be scanned to initiate transcription.

This research provides an in-depth look into the early steps of the process of converting DNA into mRNA. The biological machinery used to initiate and control this action is highly conserved between yeast and humans, suggesting that the mechanisms for controlling the activity of these factors could be similar, even if their initiation processes may differ.

such as translation efficiency or transcript stability through differences in sequence or RNA secondary structure (*Arribere and Gilbert, 2013*; *Rojas-Duran and Gilbert, 2012*; *Malabat et al., 2015*; *Sample et al., 2019*; *Cuperus et al., 2017*; *Akirtava and McManus, 2021*; *Lin et al., 2019*). Furthermore, in conjunction with activators and coactivators, the efficiency of the initiation process will also establish mRNA synthesis rates. How TSS selection is governed by these factors is not well understood for the majority of eukaryotic promoters that utilize multiple TSSs.

Transcription initiation by *Saccharomyces cerevisiae* Pol II has been the subject of extensive analysis both in vivo and in vitro, and thus provides a powerful model for system for mechanistic studies of TSS selection. TSS selection by *S. cerevisiae* Pol II occurs over a range of positions located ~40–120 bp downstream of the core promoter region. Numerous lines of evidence suggest that TSS selection by *S. cerevisiae* Pol II involves a unidirectional scanning mechanism in which the preinitiation complex (PIC) assembles at an upstream core promoter and interrogates consecutive downstream positions for usable TSSs (*Giardina and Lis, 1993*; *Kuehner and Brow, 2006*; *Hampsey, 2006*; *Fishburn et al., 2016*; *Qiu et al., 2020*). TFIIH is proposed to drive Pol II scanning through ATP-dependent DNA translocase activity (*Fishburn et al., 2016*; *Tomko et al., 2021*; *Tomko et al., 2017*; *Fishburn et al., 2015*; *Fazal et al., 2015*). An optical-tweezer-based single molecule analysis of reconstituted *S. cerevisiae* PICs indicated that an ATP/dATP-induced activity within the PIC causes shortening of the distance between upstream and downstream DNA (*Fazal et al., 2015*). This shortened distance approximates the distance downstream from TATA elements where TSSs are positioned in yeast (40–120 nt) (*Struhl, 1987*) and suggests downstream DNA movement and compaction during promoter scanning by the PIC. Separately, a magnetic tweezer-based single molecule assay suggested that an initial melted region of 6 nt (a 'bubble') is the direct consequence of TFIIH's ATPase activity (*Tomko et al., 2017*). Due to inability of magnetic tweezers to detect DNA compaction in the particular setup used, how the Pol II machinery reaches downstream TSSs, whether through generation of a large bubble or translocation of a small bubble was not clear. Nevertheless, both studies agree that an ATP-dependent

PIC activity for promoter opening is likely Ssl2 within TFIIH, which has been demonstrated as a DNA translocase within purified TFIIH in vitro (*Fishburn et al., 2015*).

In support of Ssl2/TFIIH's role in movement of downstream DNA toward the PIC, *ssl2* mutants have been identified as altering TSS selection at *ADH1* and showed genetic interactions with *sua7* (TFIIB) mutants (*Goel et al., 2012*). Specifically, the identified *ssl2* mutants shifted TSSs upstream at *ADH1*. Polar shifts in TSSs distributions have been observed in mutants within Pol II, the GTFs TFIIB and TFIIF, and the PIC cofactor Sub1 (*Kuehner and Brow, 2006*; *Qiu et al., 2020*; *Goel et al., 2012*; *Yang and Ponticelli, 2012*; *Khaperskyy et al., 2008*; *Pal et al., 2005*; *Majovski et al., 2005*; *Freire-Picos et al., 2005*; *Ghazy et al., 2004*; *Chen and Hampsey, 2004*; *Faitar et al., 2001*; *Pappas and Hampsey, 2000*; *Wu et al., 1999*; *Bangur et al., 1999*; *Pardee et al., 1998*; *Sun et al., 1996*; *Sun and Hampsey, 1995*; *Pinto et al., 1994*; *Berroteran et al., 1994*; *Pinto et al., 1992*; *Hampsey et al., 1991*; *Knaus et al., 1996*). In a promoter scanning initiation mechanism, altering initiation efficiency is predicted to alter TSS distributions in a polar fashion when initiation efficiency increases or decreases. We have recently observed polar (directional) shifts at essentially all promoters examined across the genome in yeast for tested Pol II and GTF mutants, as predicted for scanning operating universally across promoter classes (*Qiu et al., 2020*). We found that hyperactive Pol II catalytic mutants shifted TSSs' distributions upstream at promoters genome-wide, consistent with a higher probability of initiation at every TSS, and thus, initiation happening on average *earlier* in the scanning process. Conversely, hypoactive Pol II catalytic mutants shift TSS distributions downstream at promoters genome-wide, consistent with initiation happening *later* in the scanning process. Our previous data on genetic interactions between Pol II and TFIIF or TFIIB support the idea that these mutations altered initiation additively (*Jin and Kaplan, 2014*), consistent with their acting in the same pathway during scanning. Tested TFIIB mutants appeared to reduce initiation efficiency while tested TFIIF mutants appeared to increase initiation efficiency. Consistent with this idea, a double mutant between TFIIF and a hyperactive Pol II mutant had stronger effects on TSS shifts than either mutant alone across all promoters. Pol II mutants are proposed to control initiation efficiency because active site residues important for catalytic activity alter TSS distributions correlating with the strengths of their catalytic defects in RNA chain elongation. Initiation by promoter scanning should be dependent both on Pol II catalytic rate together and by whichever factors control the actual scanning step, that is, presumptively the rate and processivity of TFIIH as the putative scanning translocase. Therefore, to understand how promoter scanning works, it is critical to understand how TFIIH contributes and how its activity is regulated within the PIC.

We have described the scanning process previously using a 'Shooting Gallery' analogy (*Qiu et al., 2020*; *Kaplan, 2013*). In this model, initiation is controlled by the rate (TFIIH's translocase activity) in which a 'target' (TSS) passes the 'line of fire' (the Pol II active site) along with the rate of firing (Pol II catalytic activity) and the size of the target (innate sequence strength). Together, the cooperation and competition between these rates determines the probability a target is hit (initiation happening). Alteration of enzymatic activities supporting initiation, either the Pol II active site or TFIIH translocation, should have predictable effects on overall TSS distributions when initiation proceeds by scanning. In addition to the TFIIH translocation rate, it is predicted that processivity of TFIIH DNA translocation should strongly modulate scanning, but in ways distinct from controlling innate initiating efficiency. Here, TFIIH processivity would control the probability that a TSS could be reached during a scanning foray from a core promoter, which appears to be facilitated by Ssl2's translocase activity (*Fazal et al., 2015*). Optical tweezer experiments are consistent with TFIIH within the PIC having median processivity on the order of ~90 bp, consistent with the average distance of TSSs downstream of yeast TATA boxes. However, purified holo-TFIIH from yeast has much reduced measured processivity and human TFIIH has essentially none (*Tomko et al., 2021*). Given that TFIIH activity is predicted to be altered extensively by cofactor interactions in both transcription and nucleotide excision repair (NER), it will be important to understand TFIIH functions within the PIC in vivo. How alterations to Ssl2/TFIIH translocase activity control TSS distributions has not been extensively investigated.

To test if distinct alterations to Ssl2 function have broad effects on promoter scanning and TSS selection, we used existing and newly identified *ssl2* alleles to examine their effect on TSS distributions genome-wide. Our novel alleles were identified through use of genetic reporters we have developed and found to be sensitive to different kinds of initiation defects (*Malik et al., 2017*). We have found that *ssl2* alleles affect TSS distributions for the majority of promoters in yeast and for all promoter

classes. Furthermore, we find that *ssl2* alleles alter TSS selection distinctly from how changes to the Pol II active site alter TSS selection, consistent with a distinct role for Ssl2 in promoter scanning. *ssl2* alleles appear to extend or truncate scanning windows at promoters genome-wide, consistent with increase or decrease in the processivity of scanning. Scanning-window truncating alleles map throughout the Ssl2 structure, consistent with a hypothetical loss of functions (LOF) in Ssl2 DNA translocase enzymatic activity that lead to decreased TFIIH processivity. Conversely, scanning-window extending *ssl2* alleles are much more localized within the Ssl2 N-terminus, including conserved residues within regions that connect Ssl2 helicase domains to the TFIIH component and regulator of Ssl2 activity Tfb2. Our alleles are consistent with alteration to an Ssl2 regulatory domain resulting in modulated translocase activity or TFIIH processivity. We further test our model for initiation by promoter scanning through examination of genetic interactions of initiation-altering Pol II/GTF alleles and *ssl2* alleles. The genetic interactions between Pol II/GTF and *ssl2* alleles support the idea of two major networks controlling TSS selection in *S. cerevisiae*. One network shapes TSS distributions through affecting initiation efficiency, represented by Pol II, TFIIB, and TFIIF functions. A second network appears to alter TSS distributions through regulating TFIIH's processivity, and includes Ssl2, Sub1, and potentially TFIIF.

## Results

### Existing *ssl2* alleles show transcription-dependent growth phenotypes and distinct TSS usage patterns

To understand how TSSs are identified by promoter scanning and the potential roles for TFIIH, we first examined previously identified *ssl2* mutants (***Goel et al., 2012***; ***Gulyas and Donahue, 1992***; ***Qiu et al., 1993***; ***Lee et al., 1998***) for transcription-related phenotypes that we have demonstrated are predictive of specific initiation defects (***Figure 1A***). Two such phenotypes relate to altered initiation at the *IMD2* gene, whose promoter is regulated by a TSS switch (***Figure 1A***, *IMD2*, *imd2Δ::HIS3*; ***Kuehner and Brow, 2008***; ***Jenks et al., 2008***). We have previously shown that tested mutants that shift TSSs upstream due to altered promoter scanning result in an inability to express a functional *IMD2* transcript, causing sensitivity to the IMPDH inhibitor mycophenolic acid (MPA) (***Qiu et al., 2020***; ***Jin and Kaplan, 2014***; ***Malik et al., 2017***; ***Kaplan et al., 2012***; ***Figure 1A***, **IMD2**). In the presence of GTP starvation induced by MPA, wild-type (WT) strains shift start site usage at *IMD2* from TATA-proximal GTP-initiating TSSs to a downstream ATP-initiating TSS. This shift results in a functional *IMD2* transcript that is required for yeast to survive MPA. Catalytically hyperactive Pol II mutants (termed GOF for 'gain-of-gunction') do not shift TSS usage to the downstream functional *IMD2* TSS but instead shift to intermediately positioned non-functional upstream sites, rendering yeast sensitive to MPA (MPA$^S$) (***Malik et al., 2017***). Pol II GOF mutants with MPA$^S$ phenotypes and a *tfg2* MPA$^S$ mutant were additionally found to shift TSS distributions upstream at *ADH1* and subsequently genome-wide (***Qiu et al., 2020***; ***Kaplan et al., 2012***; ***Eichner et al., 2010***). Correlation between strength of MPA-sensitive phenotypes and quantitative upstream TSS shifts at *ADH1* and genome-wide suggest that MPA sensitivity is a strong predictor for upstream-shifting TSS mutants. Conversely, we found previously that mutants shifting TSSs downstream (reduced catalytic activity 'LOF' Pol II mutants) constitutively express *IMD2* in the absence of using MPA as inducer (***Kaplan et al., 2012***). Pol II TSS downstream shifting phenotypes at *IMD2* can be detected using a reporter allele where *IMD2* is replaced by *HIS3*, placing *HIS3* under control of *IMD2* promoter and TSS selection (***Figure 1A***, *imd2Δ::HIS3*; ***Malik et al., 2017***). Indeed, these same LOF Pol II mutants shift TSS distributions downstream at *ADH1* and genome-wide (***Qiu et al., 2020***; ***Jin and Kaplan, 2014***; ***Kaplan et al., 2012***). These previous results suggest that we have phenotypes predictive of alterations to promoter scanning in both directions and can form the basis of a system to characterize mutants across the genome for effects on TSS selection by promoter scanning.

We used site-directed mutagenesis and plasmid shuffling to recreate and phenotype five previously described *ssl2* mutants, reasoning that this would allow us a first glimpse at the effects and potential diversity present in these classic alleles. These alleles are *ssl2-rtt* (*ssl2* E556K) (***Lee et al., 1998***), *ssl2-DEAD* (*ssl2* V490A/H491D), and *SSL2-1* (*ssl2* W427L) (***Gulyas and Donahue, 1992***), *ssl2-508* (*ssl2* H508R) (***Goel et al., 2012***), and *rad25-ts24* (*ssl2* V552I/E556K) (***Qiu et al., 1993***; ***Figure 1B and C***). Analysis of these five *ssl2* mutants showed phenotypes consistent with altered TSS selection (***Figure 1C***). First, *ssl2-DEAD* and *ssl2-508* exhibited strong and weak MPA$^S$ phenotypes, respectively.

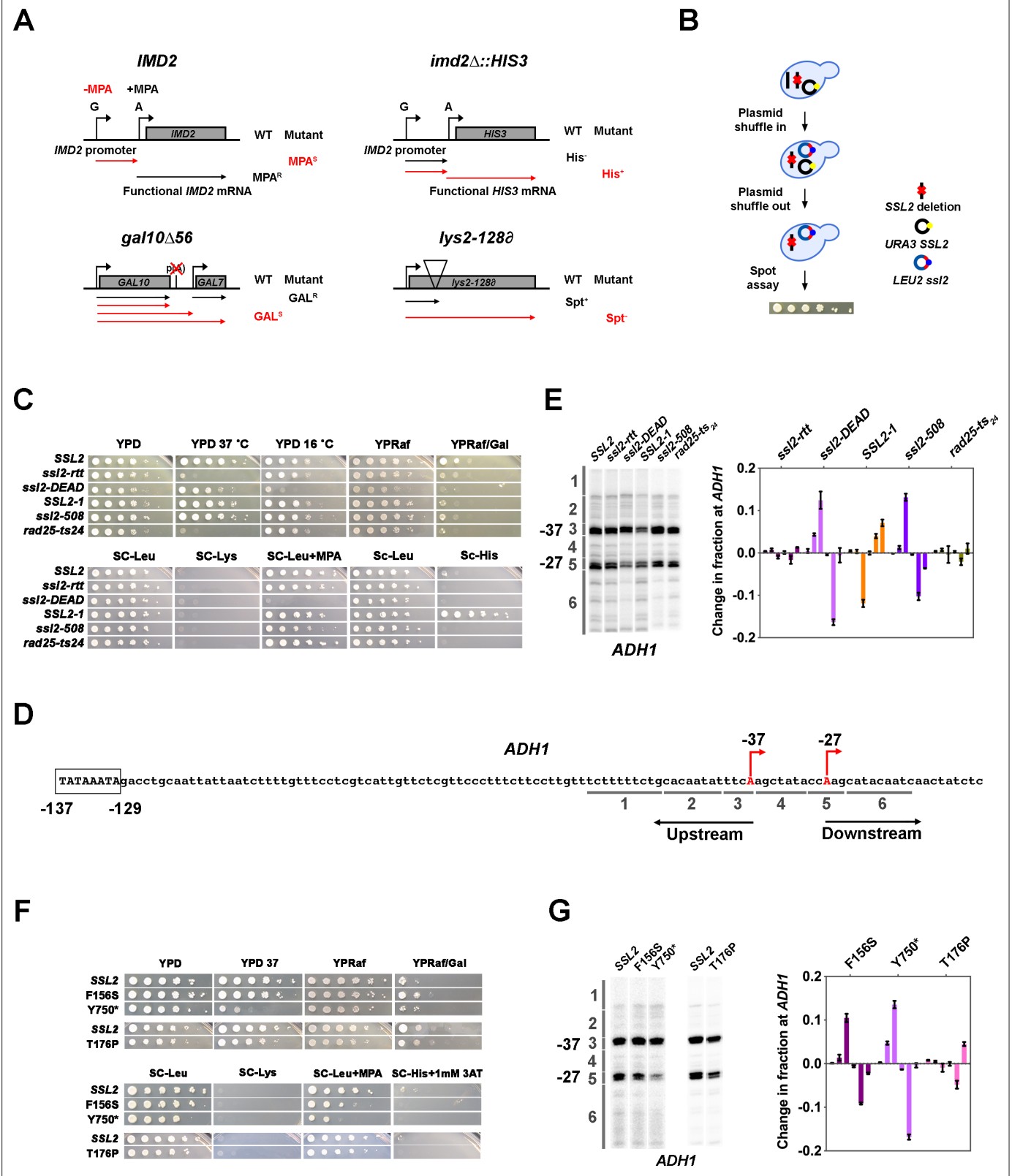

**Figure 1.** Genetic screening identifies novel *ssl2* alleles with transcriptional defects. (**A**) Schematics illustrating four transcriptional phenotypes utilized in this study (*IMD2*, *imd2Δ::HIS3*, *gal10Δ56*, *lys2-128∂*). (**IMD2**) In GTP replete conditions, *IMD2* transcription initiates at upstream transcription start sites (TSSs) utilizing GTP as the first nucleotide. These are non-functional due to the presence of a terminator prior to the *IMD2* coding sequence. Upon GTP starvation induced by the drug mycophenolic acid (MPA), initiation shifts to a downstream ATP-initiated TSS, enabling functional *IMD2* expression,

*Figure 1 continued on next page*

*Figure 1 continued*

conferring resistance to MPA. Inability to shift to the downstream *IMD2* TSS in the presence of MPA leads to MPA sensitivity (MPA$^S$), commonly found in mutants that shift TSSs upstream. (*imd2Δ::HIS3*) The *IMD2* ORF is replaced by the *HIS3* ORF creating a transcriptional fusion between the *IMD2* promoter and *HIS3*. If the downstream *IMD2* TSS is used constitutively, *HIS3* mRNA production supports growth on medium lacking histidine (SC-His). Constitutive use of the downstream *IMD2* start site by downstream TSS shifting mutants in the absence of MPA can be detected by a His$^+$ phenotype in cells with the *imd2Δ::HIS3* reporter. (*gal10Δ56*) Deletion of *GAL10* polyadenylation signal at p(A) site interferes with *GAL10* 3'-end formation and *GAL7* initiation, resulting in galactose toxicity (*Nogales and Greber, 2019*). Transcription mutants that increase *GAL10* 3'-end formation or *GAL7* initiation allow suppression of galactose toxicity and display galactose resistance (GAL$^R$). (*lys2-128∂*) Insertion of a Ty transposon ∂ element into *LYS2* causes premature termination of Pol II initiating at *LYS2*, resulting in lysine auxotrophy. Certain mutants allow expression of *LYS2* from a cryptic site within the Ty ∂ element and allow yeast growth on medium lacking lysine (SC-Lys), conferring the 'Suppressor of Ty' (Spt$^-$) phenotype. (**B**) Schematic illustrating the plasmid shuffle assay to examine *ssl2* mutant phenotypes (see Materials and methods). (**C**) Transcription-related growth phenotypes of five classical *ssl2* alleles. All spot dilutions shown throughout the figures are representative of at least two independent transformants. (**D**) Schematic illustrating the TSS region of *Saccharomyces cerevisiae ADH1*. The *ADH1* promoter contains two commonly used TSSs (red letters), which are 37 nt (–37) and 27 nt (–27) upstream from the translational start codon. For quantification of changes to *ADH1* TSS distribution by primer extension, *ADH1* TSS usage is separated into six bins. (**E**) Left panel, primer extension-detected TSS usage of the wild-type (WT) and five existing *ssl2* mutants at *ADH1*. Right panel, quantitative analysis of five classical *ssl2* alleles at *ADH1*. Average of ≥3 biological replicates ± standard deviation are shown. (**F**) Transcription-related growth phenotypes of *ssl2* alleles homologous to human disease alleles of XPB. (**G**) Primer extension detection of TSS usage at *ADH1* for alleles shown in (**F**). Average of ≥3 biological replicates ± standard deviation are shown.

The online version of this article includes the following figure supplement(s) for figure 1:

**Source data 1.** *Figure 1E* Primer extension gel (annotated).

**Source data 2.** *Figure 1G* Primer extension gel (annotated).

**Source data 3.** *Figure 1E and G* Graph data.

**Source data 4.** *Figure 1E* Primer extension gel (raw).

**Source data 5.** *Figure 1G* Primer extension gel (raw).

**Figure supplement 1 source data 1.** *Figure 1—figure supplement 1* Graph data.

**Figure supplement 1.** Growth phenotypes of human disease related and RED motif *ssl2* mutants.

These results were consistent with prior analysis of initiation at *ADH1* in these mutants (*Goel et al., 2012*) suggesting that our genetic phenotypes using *IMD2* are predictive of potentially global alterations to TSS selection. *SSL2-1* exhibited a His$^+$ phenotype, which is predictive of downstream shifts in TSS usage and consistent with its identification as a dominant mutant bypassing an inhibitory stem loop in the *his4-316* mRNA (*Gulyas and Donahue, 1992*). We now can rationalize the original suppressor of stem loop (SSL) phenotype of *SSL2-1* as usage of TSS downstream or within the inhibitory 5' stem loop *his4-316* sequence (though not apparent in *Gulyas and Donahue, 1992*). *ssl2-rtt* and *rad25-ts24* show temperature-sensitive phenotypes as expected; however, assayed transcription-related plate phenotypes were not observed. Due to the absence of His$^+$ or MPA$^S$ phenotypes, we predicted that *ssl2-rtt* or *rad25-ts$_{24}$* alleles would not shift TSS usage. Notably, there was no *lys2-128∂* Spt$^-$ phenotype observed among these five existing *ssl2* alleles, in contrast to our previous observation of an Spt$^-$ phenotype in a subset of MPA$^S$ Pol II TSS alleles (*Kaplan et al., 2008*; *Braberg et al., 2013*), our first indication that *ssl2* alleles may alter TSS selection in a distinct fashion from Pol II mutants.

To quantitatively examine TSS usage of these *ssl2* mutants, we chose *ADH1* initiation. *ADH1* has been widely used as a model gene for TSS studies in *S. cerevisiae*. It contains two major TSSs that are 27 and 37 nt upstream of the start codon (*Figure 1D*). Using primer extension, transcription products of these two TSSs appear as bands of differing mobility on denaturing polyacrylamide gels (*Figure 1—figure supplement 1A*, left panel WT). Other TSSs' positions show minor usage. In most studies, the two major starts are compared qualitatively, but such comparisons miss meaningful alterations that may tell us about initiation mechanisms. To establish that genetic phenotypes using *IMD2* correlate with altered TSS selection elsewhere in the genome, our quantitative analysis of the *ADH1* promoter divides TSSs observed by primer extension into six bins from upstream to downstream, in which bin 3 and 5 each contain the TSS for one of the major *ADH1* mRNA isoforms (*Figure 1D* and *Figure 1—figure supplement 1A*, left panel WT). In order to compare a mutant's TSS distribution to that of WT, distributions are normalized, and the WT distribution is subtracted from tested mutant distributions bin by bin (*Figure 1—figure supplement 1A*, middle and right panel). Negative or positive values indicate that the mutant has relatively lower or higher usage for TSSs in that particular bin, respectively (*Figure 1—figure supplement 1A*, right panel). For example, the Pol II GOF

allele E1103G increases relative TSS usage at upstream minor sites (TSS bin 2) and decreases relative TSS usage at the downstream major site (TSS bin 4) (*Figure 1—figure supplement 1A*, E1103G). Because of the dramatic effect of E1103G on TSS usage, the change of TSS usage can be easily visually detected on a primer extension gel (*Figure 1—figure supplement 1A*, left panel E1103G). However, less visually obvious but highly reproducible phenotypes are detected upon quantification (*Figure 1—figure supplement 1A*, right panel E1103G-WT). As predicted from plate phenotypes observed and a previous study (*Goel et al., 2012*), *ssl2-DEAD* and *ssl2-508* showed upstream shifts in their *ADH1* TSS distributions (*Figure 1C and E*). However, we observed that these *ssl2* alleles were quantitatively distinct in the amount of upstream shifting from Pol II and GTF alleles with comparable MPA sensitivities. *ssl2-DEAD* and *ssl2-508* appeared primarily to reduce downstream TSS usage (loss in bin 5 and gain in bin 3), whereas E1103G has its largest gain in bin 2 (compare *Figure 1—figure supplement 1A* E1103G, and *Figure 1E ssl2-DEAD* and *ssl2-508*). Consistent with the prediction based on its *imd2Δ::HIS3* phenotype, the His⁺ *SSL2-1* mutant shifted the overall TSS distribution downstream through increased relative downstream TSS utilization (*Figure 1C and E*). Previously, it had been concluded that *SSL2-1* had no obvious effects on TSSs distribution when comparing usage of just the two major starts (*Goel et al., 2012*). Two other mutants, *ssl2-rtt* and *rad25-ts₂₄*, had no obvious effects on TSS utilization at *ADH1*, consistent with their lack of plate phenotypes (*Figure 1C and E*).

We additionally constructed and tested human disease-related XPB mutations (*Oh et al., 2006*; *Cleaver et al., 1999*; *Weeda et al., 1997*; *Weeda et al., 1991*) in the yeast *SSL2* system, together with mutations in the ultra-conserved arginine-glutamic acid-aspartic acid (RED) motif. As human disease-related alleles are many times found in conserved residues, we reasoned that some may have effects on Ssl2 biochemistry detectable in our sensitive system. We examined four human disease-related mutants that confer distinct inherited autosomal recessive disorders xeroderma pigmentosum (XP), trichothiodystrophy (TTD), and Cockayne syndrome (CS). Of these, Q592*, which creates a C-terminally truncated Ssl2 protein (*Oh et al., 2006*), confers lethality (*Figure 1—figure supplement 1B*), and T176P (*Cleaver et al., 1999*; *Weeda et al., 1997*; *Weeda et al., 1991*) confers little if any growth defects and no MPA^S or His⁺ phenotypes (*Figure 1F*), consistent with unaltered TSS usage (*Figure 1G*). In contrast, F156S (*Cleaver et al., 1999*; *Weeda et al., 1991*; *Vermeulen et al., 1994*) conferred a mild MPA^S phenotype and shifted TSS distribution upstream at *ADH1* (*Figure 1G*). Mutant Y750*, which mimics a disease-related C-terminally truncated protein (*Goel et al., 2012*; *Sweder and Hanawalt, 1994*; *Weeda et al., 1990*), shows a mild to moderate level MPA^S phenotype (*Figure 1F*) and shifts TSS distribution upstream at *ADH1* (*Figure 1G*). The lethal phenotypes of RED motif substitutions in *ssl2* revealed their essential roles in *S. cerevisiae*. These results suggest that a subset of human disease alleles can alter TFIIH functions when placed in the yeast system.

## Novel *ssl2* mutants with transcriptional defects

Our establishment of a genetic system sensitive to *ssl2* mutant mediated initiation defects allowed us to obtain a broad set of alleles for the study of Ssl2 function in promoter scanning by large-scale genetic screens (see Materials and methods). Our genetic screening has identified at least two phenotypically distinguishable classes of *ssl2* alleles: the first class is putatively defective for the induction of the *IMD2* gene, resulting in sensitivity to MPA, the second class confers constitutive expression of *imd2Δ::HIS3*, resulting in a His⁺ phenotype (*Figure 1A*, *Figure 2—figure supplement 1*). Other transcription-related or conditional phenotypes, Spt⁻ (*Simchen et al., 1984*), suppression of *gal10Δ56* (Gal^R) (*Kaplan et al., 2005*; *Greger and Proudfoot, 1998*) or temperature sensitivity (Csm⁻, Tsm⁻), were observed in distinct subsets of alleles of the two major classes (*Figure 1A*, *Figure 2—figure supplement 1*). The Spt⁻ phenotype reporter used in our strains, *lys2-128∂*, detects activation of a TSS within a Ty1 ∂ element at the 5' end of the *LYS2* gene (*Simchen et al., 1984*). Importantly, a subset of TSS-shifting alleles show the Spt⁻ phenotype and it is useful to further classify identified *ssl2* alleles. We observed that Spt⁻ and His⁺ phenotypes were dominant for tested alleles (not shown), suggestive of possible GOF; in contrast, there were no dominant alleles found among MPA^S mutants, consistent with either recessive LOF mutations or the nature of the phenotype (sensitivity) or both. We then asked if TSS usage at *ADH1* was altered predictably in the mutant classes as we observed for existing *ssl2* mutants and previously studied Pol II and other GTF mutants. We find that the two major classes of *ssl2* mutants exhibited predicted TSS shifts, with all tested MPA^S alleles shifting TSS

usage upstream and all His⁺ alleles shifting TSS usage downstream (*Figure 2—figure supplement 2*) validating our genetic method for identifying *ssl2* alleles conferring altered initiation properties.

We examined our substitution mutants in the context of the structure of Ssl2 within the yeast PIC as determined by the Cramer lab (*Schilbach et al., 2021*) to understand how these alleles relate to PIC architecture and interactions. Substitutions causing MPAˢ phenotypes and upstream TSS shifts alter amino acids distributed across the protein, with a large number in conserved domains and highly conserved residues (*Figure 2A*, *Figure 2—figure supplement 3*). In contrast, mutations related to His⁺ phenotypes, while also generally conserved, alter amino acids clustered N-terminally, within a domain predicted (*Luo et al., 2015*; *He et al., 2016*) and now observed to be homologous to Ssl2's interaction partner Tfb2 (*Schilbach et al., 2021*; *Greber et al., 2019*; *Figure 2A*). This domain was first fully observed in a human core TFIIH structure (PDB 6NMI) (*Greber et al., 2019*) but has now been observed in just published higher resolution PIC structures (*Schilbach et al., 2021*; *Figure 2B*). This key visualization of the conserved Ssl2 N-terminus allows placement of most *ssl2* mutant residues identified as His⁺ or both His⁺ and Spt⁻ (*Figure 2C and D*). In addition, the Spt⁻ phenotype, not previously observed for classical *ssl2* alleles, was found exclusively within a subset of stronger downstream TSS shifting His⁺ mutants (*Figure 2A*). The coincidence between His⁺ and Spt⁻ phenotypes for *ssl2* mutants is in contrast to what is observed for Pol II mutants. In our previous studies of Pol II mutants, the Spt⁻ phenotype was observed in Pol II catalytic center substitutions, overlapping with MPAˢ to a large extent, and tightly linked with increased Pol II activity (*Kaplan et al., 2012*; *Kaplan et al., 2008*; *Braberg et al., 2013*). Pol II alleles with both Spt⁻ and MPAˢ phenotypes increased the efficiency of TSS usage resulting in upstream shifts to TSS distributions across promoters in vivo. However, in our identified *ssl2* alleles, none of the Spt⁻ mutants also conferred MPAˢ (*Figure 2A*). These observations together are consistent with distinct effects on structure and function in *ssl2* mutant classes.

Some TSS shifting substitutions are located on the Ssl2 surface (*Figure 2B*) and a subset (e.g. D610) are located proximal to DNA (*Figure 2C*). D610A confers an upstream TSS shift. In contrast, R636C, which is also close to DNA, confers a downstream TSS shift. F498, which is located in the groove of Ssl2 lobe 1 and facing DNA, caused an upstream TSS shift when substituted with leucine. Additionally, a small patch of residues with many TSS shifting substitutions is found on the Ssl2 lobe 1 surface (*Figure 2A*). These substitutions are from the helicase domain 1 and shift TSSs upstream, including D522V, K523N, N528I, F529L, and E537G. In addition, substitutions I383T and K372E are in residues proximal to this small patch but shift TSS downstream. Intriguingly a number of alleles of both classes are found in the Ssl2 N-terminal domain homologous to Tfb2 (TFB2C-like or 'Clutch') that forms interaction with Tfb2 and bridges Tfb2 with Ssl2 helicase domain 1 (*Figure 2A–D*). As Tfb2/p52 recruits Ssl2/XPB into TFIIH (*Jawhari et al., 2002*) and stimulates Ssl2/XPB catalytic activity (*Coin et al., 2007*), this interface is of special interest for how other factors might communicate with Ssl2/XBP function in both transcription and NER. Highlighting the uniqueness and potential plasticity of this region, we identified multiple substitutions at the conserved N230 in this domain, with N230D/S conferring MPAˢ and an upstream TSS shifts, while N230I conferred both His⁺ and Spt⁻ phenotypes and a downstream TSS shift (*Figure 2A–D*). These results suggest that altered Ssl2 DNA or intradomain interactions alter Ssl2 function in TSS selection in distinct ways, likely through distinct effects on Ssl2 biochemical activity, discussed below and in Discussion.

## *ssl2* alleles behave differently from Pol II and other GTF alleles for TSS usage

We highlight two alleles as representative of distinct *ssl2* allele classes in comparison to examples of the two Pol II allele classes: *ssl2* N230D, which is MPA sensitive and appears to reduce ability to use downstream TSSs at *ADH1*, and *ssl2* N230I, which shows both His⁺ and Spt⁻ phenotypes and shifts TSSs downstream relative to WT at *ADH1* (*Figure 2E and F rpb1* alleles, 2G and H *ssl2* alleles). As we have shown previously, *rpb1* mutants also fall into two major classes regarding transcription phenotypes and TSS shifts. As a comparison, *rpb1* E1103G is MPAˢ and shifts TSS usage upstream while *rpb1* H1085Y is His⁺ and shifts TSS usage downstream (*Malik et al., 2017*; *Kaplan et al., 2012*). MPA sensitivities for both *rpb1* E1103G and *ssl2* N230D correlate with an upstream TSS shift at *ADH1*, similarly to our observation for existing *ssl2* MPAˢ alleles, *ssl2-DEAD* and *ssl2-508*. However, all *ssl2* MPAˢ alleles examined appear to shift TSS distribution upstream by limiting or truncating TSS usage at downstream sites and not by activating lowly used upstream sites as *rpb1* E1103G does (and as known

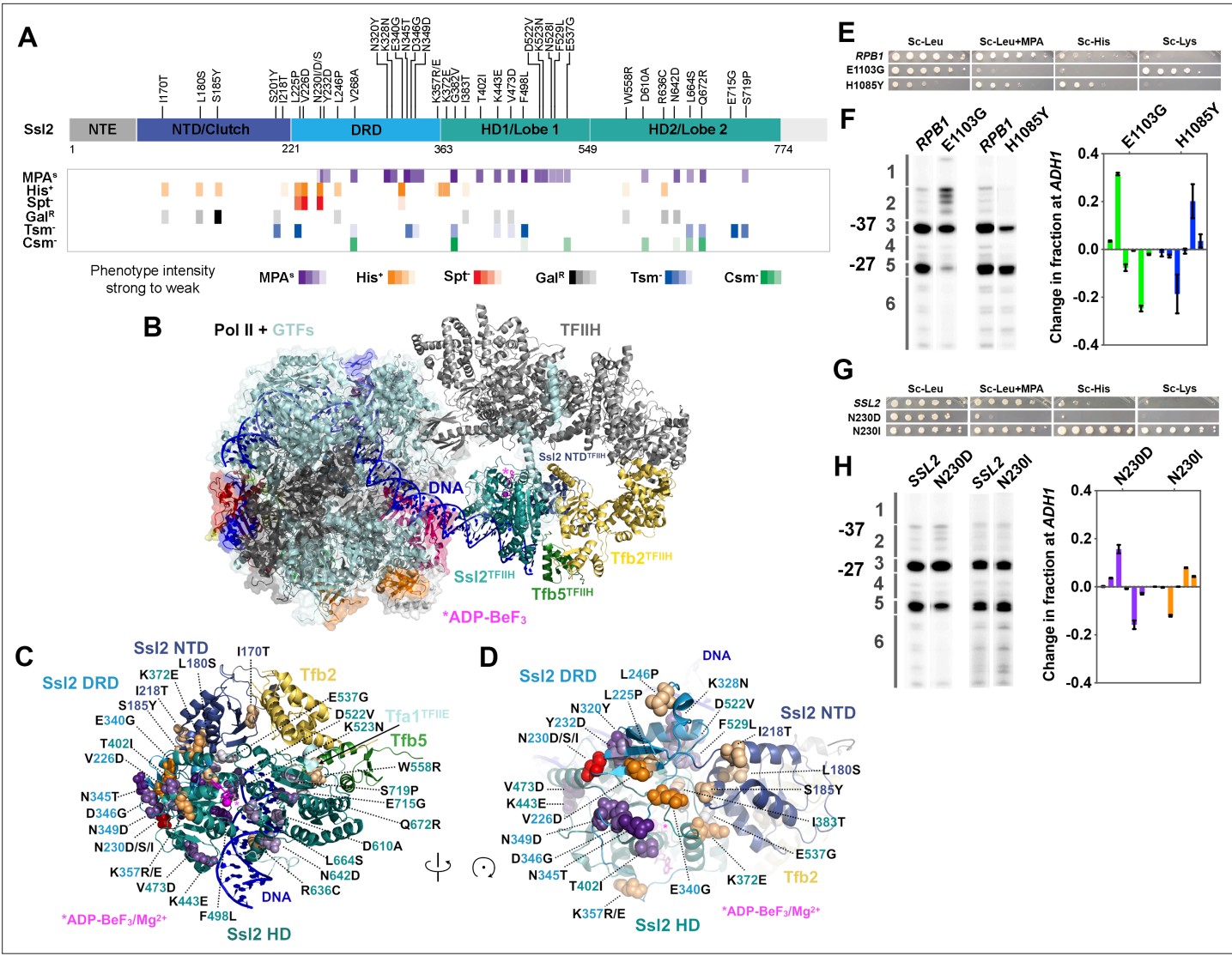

**Figure 2.** *ssl2* alleles have distinct behavior from polymerase II (Pol II) and other general transcription factors (GTFs) alleles for transcription start site (TSS) usage. (**A**) Identified *ssl2* substitutions and their phenotypes relative to Ssl2 domains. Structure colored by identified Ssl2 domains as in *Greber et al., 2019*. Light gray areas in the schematic have not yet been observed in any Ssl2 or homolog structures. (**B**) Position of Ssl2 relative to TFIIH and Pol II in the yeast PIC (PDB 7O4I). (**C**) Identified substituted residues illustrated as spheres on cartoon of the Ssl2 structure from (**B**). Residue numbers are color-coded by Ssl2 domain color from (**A**). (**D**) Rotation of (**C**) illustrating the mutants clustered at NTD/DRD/helicase domain (HD) one interface. (**E**) Spot assay showing example *rpb1* mutant transcription phenotypes. (**F**) Primer extension and quantification showing *rpb1* mutant effects on *ADH1* TSS distribution. Color coding of *rpb1* allele class in this graph is used throughout the figures. Green bars represent upstream shifting *rpb1* alleles when used to annotate figures. Blue bars represent downstream shifting *rpb1* alleles. Averages of ≥3 biological replicates ± standard deviation are shown. (**G**) Spot assay showing example *ssl2* mutant transcription phenotypes. (**H**) Primer extension and quantification showing example *ssl2* mutant effects on *ADH1* TSS distribution. Color coding of *ssl2* allele class in this graph is used throughout the figures. Orange bars represent downstream shifting *ssl2* alleles when used to annotate figures. Purple bars represent upstream shifting *ssl2* alleles. Averages of ≥3 biological replicates ± standard deviation are shown.

The online version of this article includes the following figure supplement(s) for figure 2:

**Source data 1.** *Figure 2F and H* Graph data.

**Figure supplement 1.** Growth phenotypes of *ssl2* mutants.

**Figure supplement 2.** Transcription start site (TSS) usage of representative *ssl2* single mutants.

**Figure supplement 2—source data 1.** *Figure 2—figure supplement 2A* Primer extension gel (annotated).

**Figure supplement 2—source data 2.** *Figure 2—figure supplement 2A* Primer extension gel (annotated).

**Figure supplement 2—source data 3.** *Figure 2—figure supplement 2B* Graph data.

*Figure 2 continued on next page*

*Figure 2 continued*

**Figure supplement 2—source data 4.** *Figure 2—figure supplement 2A* Primer extension gel (raw).

**Figure supplement 2—source data 5.** *Figure 2—figure supplement 2A* Primer extension gel (raw).

**Figure supplement 3.** Alignment of Ssl2 homologs illustrating position of substitutions and the conservation of affected residues.

TFIIF alleles do) (*Fishburn et al., 2015*; *Figure 2F and H*). This behavior suggests that *rpb1* and *ssl2* alleles may alter TSS usage by distinct mechanisms.

## TSS sequencing identifies global effects on TSSs in *ssl2 alleles* in *S. cerevisiae*

To gain an insight into the impact of TFIIH's activity on TSSs genome-wide, we have examined 5′ ends of RNA transcripts for these two classes of *ssl2* allele in *S. cerevisiae* by performing TSS sequencing (TSS-seq) (*Qiu et al., 2020*; *Vvedenskaya et al., 2015*; *Figure 3A*). In total, six *ssl2* alleles were analyzed along with a WT control, including three His+ alleles (L225P, N230I, and R636C) that shift TSSs

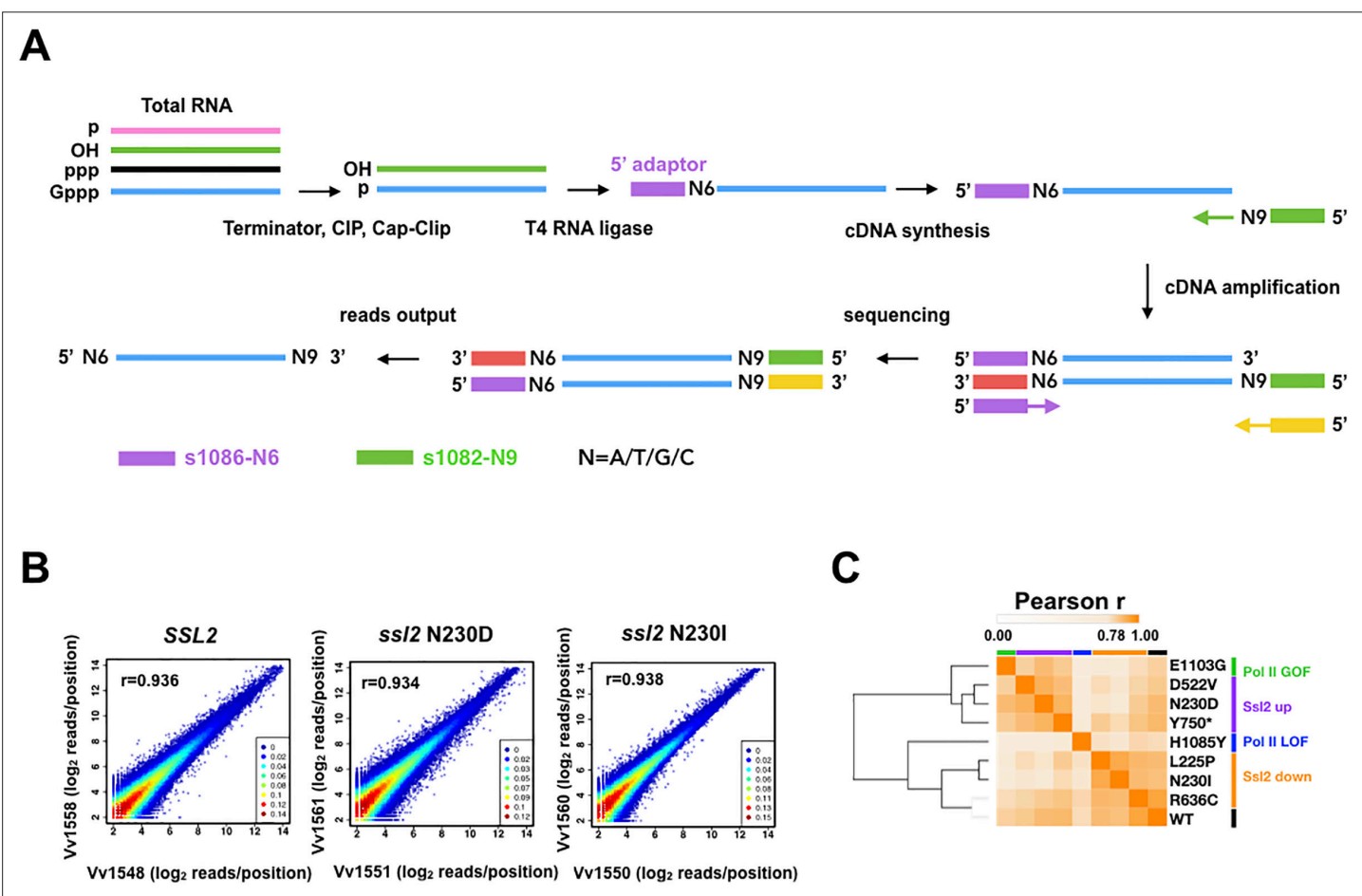

**Figure 3.** Transcription start site sequencing (TSS-seq) identifies global *ssl2* initiation effects. (**A**) TSS-seq library construction as in *Vvedenskaya et al., 2015*. See Materials and methods for details. (**B**) Scatter plot showing the correlation of log₂ transformed reads at individual genome positions for all positions ≥ 3 reads for each library for example replicate pairs for *SSL2* wild type (WT), *ssl2* N230D, or *ssl2* N230I, see Figure supplements for other libraries and description of biological replicates performed for all genotypes. (**C**) Hierarchical clustering of Pearson r correlation coefficients for libraries of combined biological replicates.

The online version of this article includes the following figure supplement(s) for figure 3:

**Source data 1.** *Figure 3C* Heatmap data.

**Figure supplement 1.** Correlation of read counts between transcription start site sequencing (TSS-seq) replicates at TSSs across the genome.

**Figure supplement 1—source data 1.** *Figure 3—figure supplement 1B* Heatmap data.

downstream at *ADH1*, and three MPA$^S$ alleles (N230D, D522V, and Y750*) that shift TSSs upstream at *ADH1*. Furthermore, we performed TSS-seq on previously analyzed Pol II WT, E1103G, and H1085Y for direct comparison purposes using our updated protocol (Materials and methods). The positions and counts of the 5′ ends of uniquely mapped reads were extracted to evaluate correlation and assess the reproducibility between biological replicates (*Figure 3B*, *Figure 3—figure supplement 1A*). We previously found that clustering of correlation coefficients among libraries could distinguish Pol II mutants into GOF and LOF groups (*Qiu et al., 2020*). Here, we also found that mutant phenotypic classes were distinguished by this clustering with *ssl2* and Pol II alleles separated suggesting effects observed at individual promoters are predictive of effects observed across the genome (*Figure 3C*, *Figure 3—figure supplement 1B*). *ssl2* alleles shift TSS distribution genome-wide.

As in *Qiu et al., 2020*, we focused on initiation within a cohort of 5979 promoters for a large number of mRNA genes and non-coding RNAs. These promoters are separated into Taf1-enriched or -depleted subclasses as a proxy for the two primary promoter classes in yeast (*Rhee and Pugh, 2012*). We took a simple approach to examine how mutants affect observed TSS distributions using a few key metrics, for example, the position in a promoter window containing the median read in the distribution as a measure of where the TSS distribution is in a particular promoter window (*Figure 4A*). We observed decreased TSS signal downstream of the WT median TSS signal in *ssl2* N230D and other *ssl2* MPA$^S$ alleles (*Figure 4B*, *Figure 4—figure supplement 1*). In contrast, substantially increased TSS usage was observed at downstream sites in *ssl2* N230I and other *ssl2* His$^+$ alleles (*Figure 4B*, *Figure 4—figure supplement 1*). To quantify the change of TSS usage, we measured the TSS shift between WT and mutant strains by subtracting the WT median TSS position within each promoter window from the mutant median TSS position (*Figure 4C*). For each mutant, we show the median TSS shifts across promoters in both heatmap and boxplot format (*Figure 4D–E*, *Figure 4—figure supplement 2A*). As predicted from our model, *ssl2* MPA$^S$ alleles shift the median TSS positions upstream at most promoters, showing a similar profile to Pol II GOF mutant (*Figure 4D*, *Figure 4—figure supplement 2A*). Also as predicted from our model, *ssl2* His$^+$ alleles shift median TSS positions downstream at the majority of promoters and show a similar profile to a Pol II LOF mutant (*Qiu et al., 2020*; *Figure 4D*, *Figure 4—figure supplement 2A*). Additionally, mutants were clustered into upstream and downstream classes based on shifts in promoter median TSS position (*Figure 4D*, *Figure 4—figure supplement 2A*). Principal component analysis (PCA) of TSS shifts distinguishes *ssl2* and Pol II mutants into four major classes, namely Pol II LOF (downstream shifting) and GOF (upstream shifting), and *ssl2* upstream and downstream shifting mutants (*Figure 4—figure supplement 2B*). We observed that the magnitude of TSS shift was consistent with the strengths of putative TSS shift-dependent growth phenotypes in *ssl2* upstream shifting mutants (*Figure 4E*). For example, alleles of N230D, D522V, and Y750*, from the left to right, show a gradient of MPA$^S$ phenotypes in genetic tests (*Figure 2A*), while also showing a gradient in TSS shift magnitudes across promoters (*Figure 4E*). Notably, the extents of TSS shifts in *ssl2* alleles are less than for Pol II activity mutants, indicating a more dramatic effect of Pol II's catalytic activity on TSS distributions (*Figure 4E*). These results are consistent with phenotypes of mutants being predictive of global TSS defects among mutants for a particular gene but not necessarily between genes, which we discuss later as indicative of different mechanisms for Pol II and Ssl2 effects on TSS selection.

## Distinct alterations to TSS distribution in *ssl2* mutants

To evaluate the effects of *ssl2* and Pol II alleles on scanning distance, the width between positions of the 10th and 90th percentiles of the TSS signal at each promoter window was determined (TSS 'spread') (*Figure 5A*). We observed obvious narrowing of TSS spreads in *ssl2* upstream shifting mutants and widening of TSS spreads in *ssl2* downstream shifting mutants (*Figure 5B–D*). The profiles of the TSS spread difference between mutant and WT at each individual promoter (*Figure 5A*) also differentiates mutants into clear shift classes (*Figure 5C*; *Figure 5—figure supplement 1A, B*). Changes in TSS spread for *ssl2* alleles are distinct from how Pol II mutants alter TSS spread (*Figure 5D*). Both classes of *ssl2* mutant show large bias in direction of shift in spread, relative to WT across a number assessments (*Figure 5—figure supplement 2D-F*). Consistently, MPA$^S$ *ssl2* alleles (those that shift TSSs upstream) showed narrowing in TSS spread at the majority of promoters while His$^+$ *ssl2* alleles (those that shift TSSs downstream) showed widening in TSS spread at the majority of promoters. These results extend the idea that while classes of initiation allele may be easily identified for Pol II or *SSL2* using the same

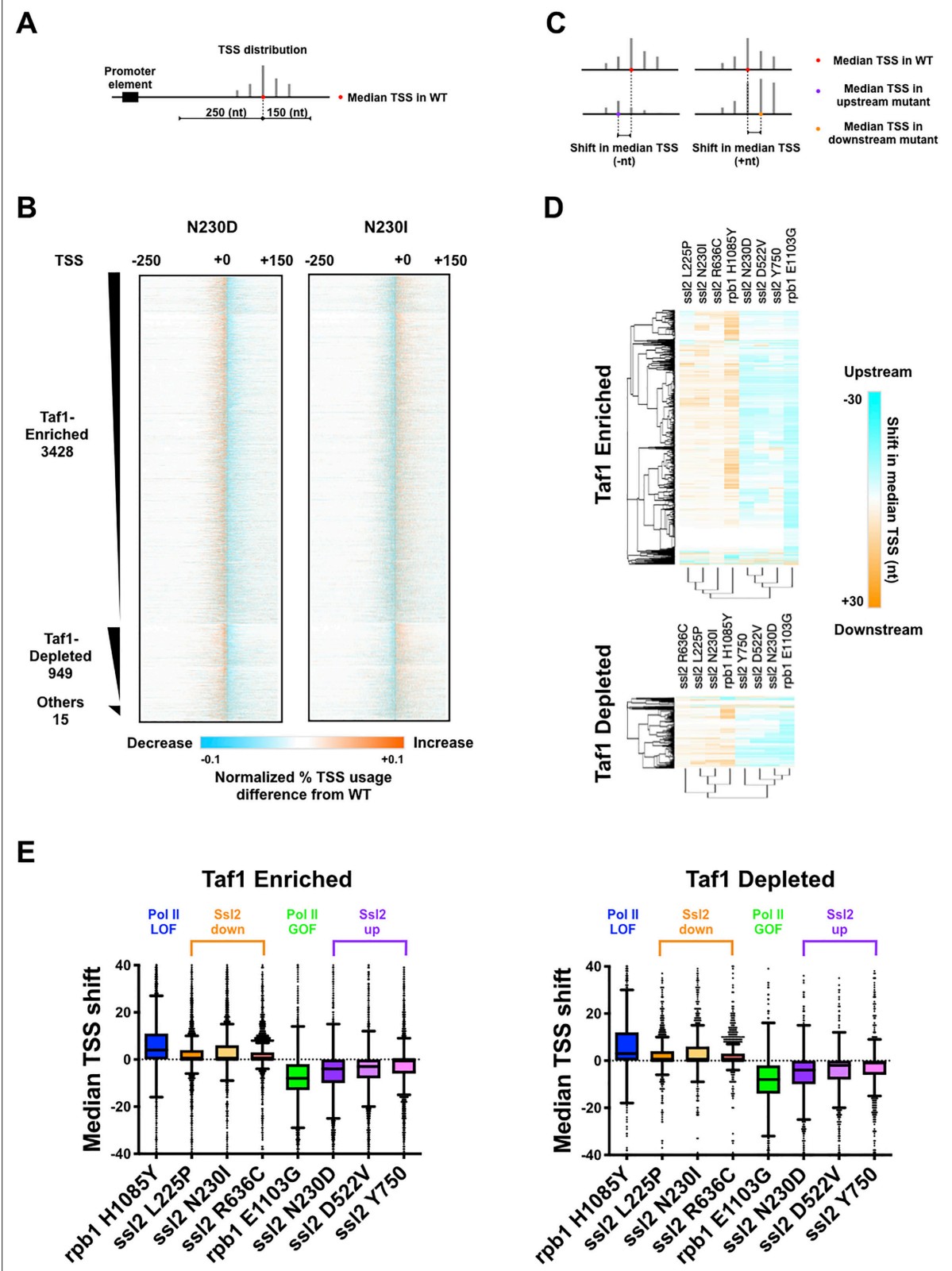

**Figure 4.** *ssl2* alleles shift transcription start site (TSS) distribution genome-wide. (**A**) Schematic indicating TSS distribution within a promoter window defined by median of the wild-type (WT) TSS distribution. (**B**) Heatmaps of TSS normalized read differences between WT and *ssl2* mutants within defined promoter windows. Promoter classes defined by enrichment or depletion of Taf1 were rank-ordered according to total reads in WT. The promoter-normalized read differences between mutant and WT are shown by a color scheme ranging from cyan (negative)-orange (positive). (**C**)

*Figure 4 continued on next page*

*Figure 4 continued*

Schematic illustrating median TSS upstream or downstream shift in a mutant relative to WT. (**D**) Heatmap and hierarchical clustering of median TSS shifts for *ssl2* or *rpb1* mutants for Taf1-enriched or -depleted promoter classes (promoters as in (**B**)). The shift of median TSS is indicated by cyan (upstream) or orange (downstream). (**E**) TSS shifts in *ssl2* mutants are less strong than in Pol II mutants. Median TSS shifts across promoters, regardless of promoter class are statistically distinguished from zero at p < 0.0001 for all mutants (Wilcoxon signed rank test).

The online version of this article includes the following figure supplement(s) for figure 4:

**Source data 1.** *Figure 4B ssl2* N230D heatmap data.

**Source data 2.** *Figure 4B ssl2* N230I heatmap data.

**Source data 3.** *Figure 4DE* Taf1-enriched heatmap and graph data.

**Source data 4.** *Figure 4DE* Taf1-depleted heatmap and graph data.

**Figure supplement 1.** Analysis of transcription start site (TSS) shifts in *ssl2* and polymerase II (Pol II) mutants.

**Figure supplement 1—source data 1.** *Figure 4—figure supplement 1 rpb1* E1103G heatmap data.

**Figure supplement 1—source data 2.** *Figure 4—figure supplement 1 rpb1* H1085Y heatmap data.

**Figure supplement 1—source data 3.** *Figure 4—figure supplement 1 ssl2* L225P heatmap data.

**Figure supplement 1—source data 4.** *Figure 4—figure supplement 1 ssl2* D522V heatmap data.

**Figure supplement 1—source data 5.** *Figure 4—figure supplement 1 ssl2* R636C heatmap data.

**Figure supplement 1—source data 6.** *Figure 4—figure supplement 1* ssl2 Y750* heatmap data.

**Figure supplement 2.** Analysis of transcription start site (TSS) shifts in *ssl2* and polymerase II (Pol II) mutants.

**Figure supplement 2—source data 1.** *Figure 4—figure supplement 2A* Heatmap data.

genetic phenotypes, their effects on TSSs at individual promoters are quantitatively and likely qualitatively distinct.

## Genetic interactions between initiation factors and *ssl2* alleles suggest distinct roles for Ssl2 and other factors in TSS scanning

To explain the observed quantitative differences between Pol II initiation *efficiency* alleles and *ssl2* alleles, we hypothesize that *ssl2* alleles that narrow TSS spreads (*ssl2* N230D and similar alleles), resulting in upstream shifts in TSS distributions, are defective in scanning *processivity* due to decreased Ssl2/TFIIH translocase activity. In contrast, *ssl2* alleles that show increased TSS spreads and usage at downstream sites (N230I and similar alleles) behave as increased scanning processivity (GOF) alleles, due to an increase in Ssl2/TFIIH activity. Alternatively, *ssl2* N230I might instead be a *scanning rate* GOF allele that decreases initiation efficiency across TSSs by decreasing the exposure time of each TSS within Pol II active site. As a consequence, there would be, hypothetically, increased TSS usage at downstream sites due to increased Pol II flux reaching those positions, similar to Pol II LOF efficiency alleles. To probe mechanisms of Ssl2 function, we designed *ssl2* genetic interaction studies for which we have specific predictions based on their possible roles (*Figure 6—figure supplement 1*). These studies emerge from our prior observations of genetic interactions from three angles between initiation factors such as Pol II activity mutants and alleles of GTFs (*Jin and Kaplan, 2014*). First, we have examined general effects on growth between classes of mutant, which can manifest as synthetic sickness or lethality, suppression, or lack of effects on growth. Second, we have examined suppression or enhancement of transcription-related phenotypes in double mutants, allowing detection of additive or epistatic interactions based on putative transcription defects at the genetic reporter loci. Finally, we have quantitatively examined effects of double mutants on TSS distributions at *ADH1*, allowing additive or epistatic interactions to be observed at the individual promoter level. In our previous studies, we found that Pol II activity mutants and alleles of GTFs (*sua7*/TFIIB and *tfg2*/TFIIF) showed additivity/suppression for transcription-dependent phenotypes as well as additivity/suppression for alterations to TSS distributions at *ADH1*. These studies suggested that these GTF alleles were functioning in same pathway as Pol II catalytic mutants, namely controlling the efficiency of initiation across individual TSSs. In order to understand how *ssl2* alleles interact with other initiation factors and how Ssl2 functions within the network of activities that are essential for TSS selection, we generated double mutants among a collection of *ssl2* alleles, Pol II activity-altering alleles, *sua7-1*, *tfg2Δ146–180*, and *sub1Δ*. To streamline display a large number of genetic interactions and growth phenotypes, we have

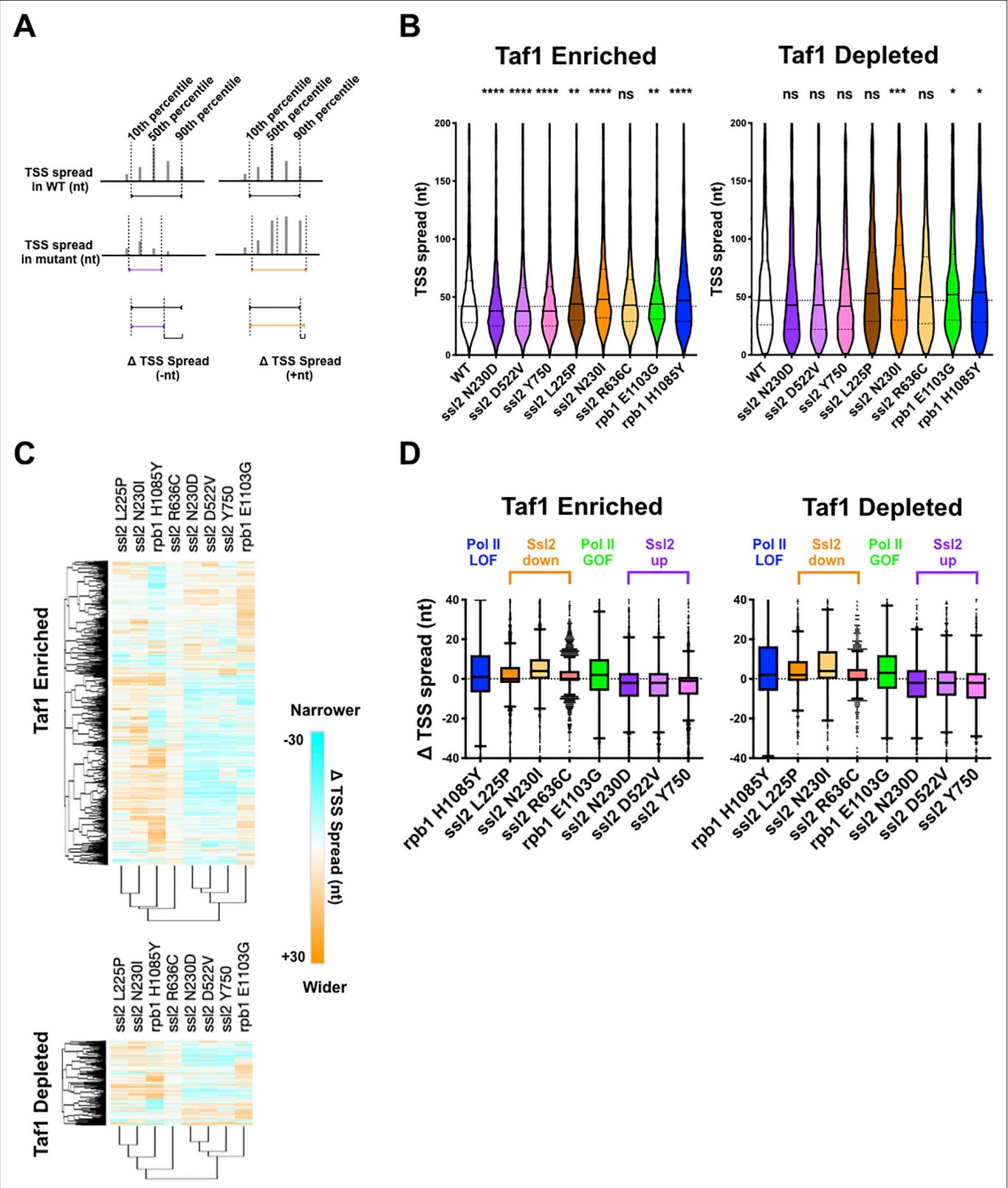

**Figure 5.** Distinct alterations to transcription start site (TSS) distribution in *ssl2* mutants. (**A**) Schematic illustrating TSS 'spread' reflecting the distance encompassing 80 % of TSSs in a promoter window and the measurement of mutant changes in TSS spread. (**B**) TSS spreads in *ssl2* and Pol II mutants at Taf1-enriched or -depleted promoters. All mutants show a statistical difference in medians from wild type (WT) at p < 0.05 (Kruskal-Wallis test with Dunn's multiple comparisons test). Asterisks indicate differences in means from WT (*p < 0.05, **p < 0.01, ***p < 0.005, ****p < 0.001). (**C**) Heatmap

*Figure 5 continued on next page*

*Figure 5 continued*

showing TSS spread changes for *ssl2* or *rpb1* mutants by promoter class (hierarchical clustering by mutant on the *x*-axis and promoter on the *y*-axis). (**D**) *ssl2* upstream and downstream shifting mutants narrow and widen TSS distributions at promoters as measured in (**A**). The median TSS spread changes of all the *ssl2* and Pol II mutants are statistically distinguished from zero at p < 0.0001 (Wilcoxon signed rank test).

The online version of this article includes the following figure supplement(s) for figure 5:

**Source data 1.** *Figure 5B* Taf1-enriched graph data.

**Source data 2.** *Figure 5B* Taf1-depleted graph data.

**Source data 3.** *Figure 5CD* Taf1-enriched heatmap and graph data.

**Source data 4.** *Figure 5CD* Taf1-depleted heatmap and graph data.

**Figure supplement 1.** Analysis of transcription start site (TSS) spread in *ssl2* and polymerase II (Pol II) mutants.

**Figure supplement 1—source data 1.** *Figure 5—figure supplement 1A* Heatmap data.

**Figure supplement 2.** Number of promoters affected by *ssl2* or polymerase II (Pol II) mutants.

**Figure supplement 2—source data 1.** *Figure 5—figure supplement 2A* Graph data.

**Figure supplement 2—source data 2.** *Figure 5—figure supplement 2B* Data table.

**Figure supplement 2—source data 3.** *Figure 5—figure supplement 2B* Graph data.

**Figure supplement 2—source data 4.** *Figure 5—figure supplement 2C* Data table.

**Figure supplement 2—source data 5.** *Figure 5—figure supplement 2C* Graph data.

**Figure supplement 2—source data 6.** *Figure 5—figure supplement 2D* Graph data.

**Figure supplement 2—source data 7.** *Figure 5—figure supplement 2E* Data table.

**Figure supplement 2—source data 8.** *Figure 5—figure supplement 2E* Graph data.

**Figure supplement 2—source data 9.** *Figure 5—figure supplement 2F* Data table.

**Figure supplement 2—source data 10.** *Figure 5—figure supplement 2F* Graph data.

converted general growth on plates and growth on phenotype-specific media to qualitative scores (*Figure 6D–F*, *Figure 6—figure supplement 2*) and these are shown as heatmaps (*Figures 6 and 7*) with primary data shown in *Figure 6—figure supplement 3*, *Figure 7—figure supplement 1*, and *Figure 7—figure supplement 2*.

Our detailed studies are summarized as follows (for detailed discussion, see version 1 of the pre-print of this work *Zhao et al., 2021*). In contrast to the suppressive/additive interactions that were broadly observed between Pol II and TFIIB/TFIIF alleles (*Jin and Kaplan, 2014*), we observe primarily epistatic effects between Pol II and *ssl2* alleles. First, the broad synthetic lethality or enhancement/additivity of transcriptional phenotypes through combining Pol II and TFIIB/TFIIF alleles that shift TSS distributions in the same direction were not observed between Pol II and *ssl2* alleles (*Figure 6*). Examples of epistatic interactions, where double mutants between Pol II and *ssl2* alleles have phenotypes of either the Pol II single mutant or the *ssl2* single mutant, were found in a number of cases. Each case supports a model where *ssl2* alleles are functioning through scanning processivity and not initiation efficiency directly. This epistasis is best reflected by nearly complete absence of synthetic lethality between *ssl2* downstream shifting alleles and Pol II downstream shifting alleles (*Figure 6A–C*), in contrast to interactions between all other classes of downstream shifting allele (e.g. Pol II, *sua7*/TFIIB, *sub1Δ*; *Jin and Kaplan, 2014*). Epistasis for both transcription phenotypes and *ADH1* TSS shifts was also observed between Pol II upstream shifting alleles and both classes of *ssl2* allele, meaning that double mutants had phenotypes of Pol II single mutants (*Figure 6A–K*, *Figure 6—figure supplement 3*). For each of these cases, results support a model where if initiation is efficient enough or early enough in a scanning window, that is, due to increased Pol II initiation activity, then increase in scanning processivity (e.g. *ssl2* N230I) loses ability to alter TSS distributions, while a decrease in scanning processivity is buffered against due to high enough gain in transcription efficiency in tested Pol II alleles. Similarly, both classes of *ssl2* alleles appeared epistatic or non-additive with Pol II downstream shifting alleles (*Figure 6A–C*, *Figure 6—figure supplement 3*), also consistent with determination of scanning window by *ssl2* activity to be upstream of ability of Pol II mutants to alter TSS distributions through altered initiation efficiency.

Interactions between *ssl2* alleles and other GTFs or *sub1Δ* reveal complexities that are of special note as they suggest non-obvious roles/interactions between these factors and Ssl2 function (*Figure 7*,

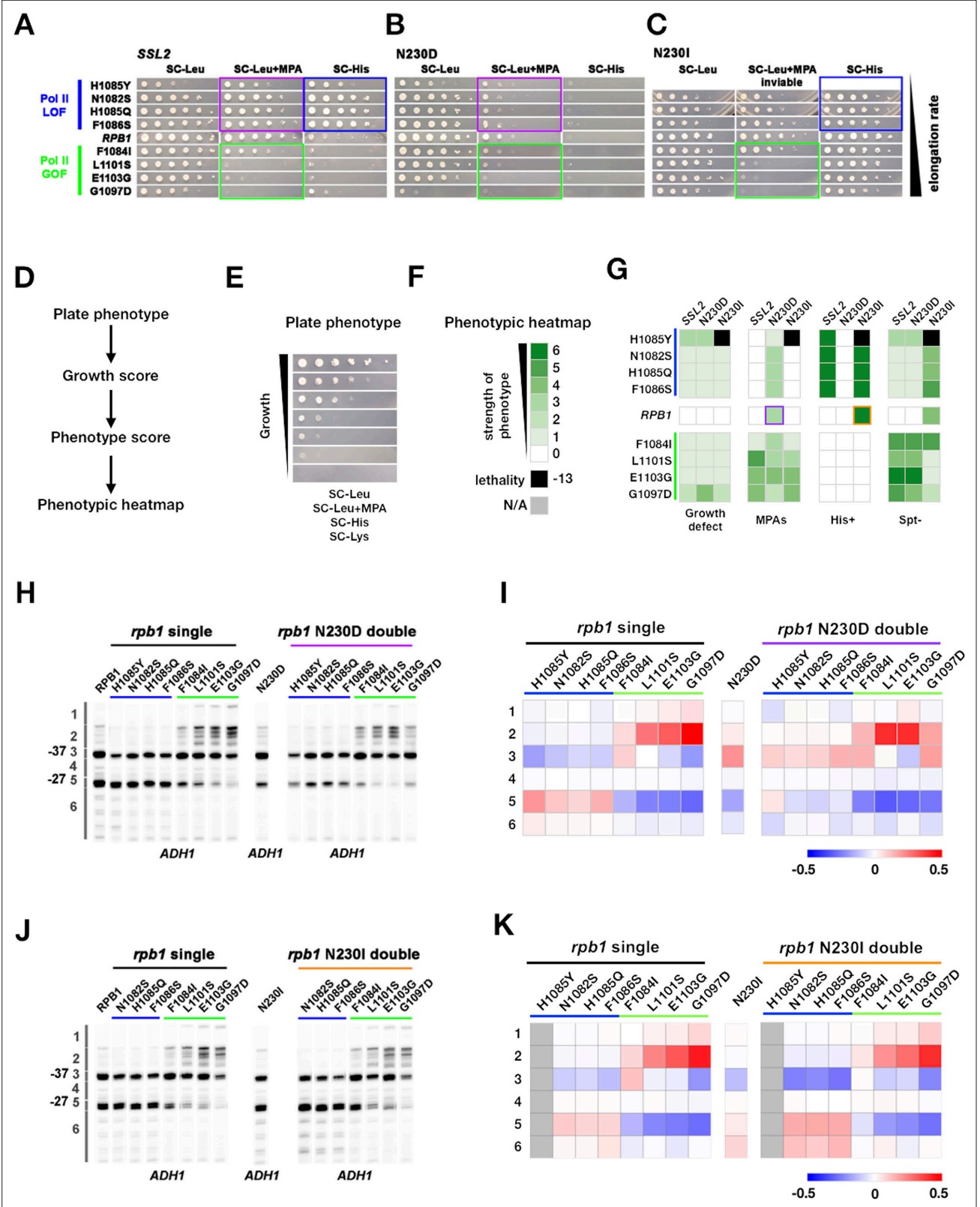

**Figure 6.** Genetic interactions between *ssl2* and polymerase II (Pol II) initiation alleles suggest distinct functions of each in initiation by scanning. (**A**) Growth phenotypes of *rpb1*, *ssl2* N230D, *ssl2* N230I single or double mutants. *rpb1* mutants represent known catalytically hyperactive alleles or genetically similar (G1097D, E1103G, L1101S, F1084I) and four with reduced catalytic activity (F1086S, H1085Q, N1082S, H1085Y). Strains are arranged according to measured Pol II elongation rate in vitro (slowest at top). (**B**) *ssl2* mycophenolic acid (MPA)-sensitive alleles are epistatic to Pol II LOF

*Figure 6 continued on next page*

*Figure 6 continued*

alleles' His$^+$ phenotypes (double mutants retain MPA$^S$ of *ssl2* single mutant while His$^+$ phenotypes of *rpb1* mutants are suppressed). Conversely, Pol II transcription start site (TSS) upstream shifting alleles appear epistatic/non-additive with *ssl2* MPA$^S$ alleles and do not show synthetic growth phenotypes. (**C**) Pol II upstream TSS shifting alleles appear epistatic to *ssl2* N230I phenotypes (MPA$^S$ retained and His$^+$ suppressed in double mutants). There are only minor synthetic defects between *ssl2* N230I and Pol II downstream TSS shifting mutants suggesting lack of synergistic defect and either mild additivity or epistasis. (Double mutant of N230I and H108Y is nearly dead and was not tested here or in E.) (**D,E**) Schematic (**D**) indicating how qualitative growth data of mutants encoded (**E**) for visualization in heatmaps. (**F**) Phenotyping heatmap legend. (**G**) Qualitative heatmaps for *ssl2* and *rpb1* genetic interactions. Growth phenotypes are detected using reporters described in ***Figure 1***. (**H**) Primer extension of *ssl2* N230D and *rpb1* mutants at *ADH1*. *ssl2* N230D appears to truncate distribution of TSSs on downstream side and is epistatic to downstream shifting *rpb1* alleles (blue bar) while upstream shifting *rpb1* alleles (green bar) are non-additive or epistatic to *ssl2* N230D. Numbered regions indicate TSS positions that were binned for quantification in (**I**). Representative primer extension of ≥3 independent biological replicates is shown. (**I**) Quantification of (**H**) with heatmap showing relative differences in TSS distribution binned by position (bins are numbered and shown in **H**). Mean changes of ≥3 independent biological replicates are shown in the heatmap. (**J**) Primer extension of *ssl2* N230I and *rpb1* mutants at *ADH1*. *ssl2* N230I appears to enhance usage of downstream TSSs and is additive with downstream shifting *rpb1* alleles (blue bar) while upstream shifting *rpb1* alleles (green bar) are epistatic to *ssl2* N230I. Numbered regions indicate TSS positions that are binned for quantification in (**K**). Representative primer extension of ≥3 independent biological replicates is shown. (**K**) Quantification of (**J**) with heatmap showing relative differences in TSS distribution binned by position (bins are numbered and shown in (**J**)). Mean changes of ≥3 independent biological replicates are shown in the heatmap.

The online version of this article includes the following figure supplement(s) for figure 6:

**Source data 1.** *Figure 6G* Heatmap data.

**Source data 2.** *Figure 6H* Primer extension gel (annotated).

**Source data 3.** *Figure 6H* Primer extension gel (annotated).

**Source data 4.** *Figure 6I rpb1* single heatmap data.

**Source data 5.** *Figure 6I* N230D single heatmap data.

**Source data 6.** *Figure 6I* N230D double heatmap data.

**Source data 7.** *Figure 6J* Primer extension gel (annotated).

**Source data 8.** *Figure 6J* Primer extension gel (annotated).

**Source data 9.** *Figure 6K rpb1* single heatmap data.

**Source data 10.** *Figure 6K* N230I single heatmap data.

**Source data 11.** *Figure 6K* N230I double heatmap data.

**Source data 12.** *Figure 6H* Primer extension gel (raw).

**Source data 13.** *Figure 6H* Primer extension gel (raw).

**Source data 14.** *Figure 6J* Primer extension gel (raw).

**Source data 15.** *Figure 6J* Primer extension gel (raw).

**Figure supplement 1.** Design of *ssl2* genetic interaction tests with efficiency alleles.

**Figure supplement 2.** The scoring method used to quantify yeast growth phenotypes and make phenotypic heatmaps.

**Figure supplement 3.** Polymerase II (Pol II) efficiency alleles are able to increase transcription start site (TSS) efficiency within the processivity defined scanning window.

**Figure supplement 3—source data 1.** *Figure 6—figure supplement 3A* Graph data.

**Figure supplement 3—source data 2.** *Figure 6—figure supplement 3B* Graph data.

---

*Figure 7—figure supplements 1 and 2*). We anticipated that *sua7-1* and *tfg2Δ146–180*, encoding mutant forms of TFIIB and TFIIF respectively, would behave strictly as TSS efficiency alleles due to their additive behavior with Pol II alleles (***Jin and Kaplan, 2014***), and therefore would similarly show epistatic effects with *ssl2* alleles. Notably, lethal phenotypes were observed between individual *ssl2* alleles and *sua7-1* or *tfg2Δ146–180* alleles for combinations between single mutants that alter TSS distributions in the same direction, distinct from their interactions with Pol II alleles (***Figure 7A***). We suggest two possibilities for this observation: first, *sua7-1* and *tfg2Δ146–180* could confer additional defects causing increased sensitivity to *ssl2* defects, for example, altered PIC integrity; second, *sua7-1* and *tfg2Δ146–180* might be sensitized to increased Ssl2 processivity (for *sua7-1*) or decreased Ssl2 processivity (for *tfg2Δ146–180*) in addition to their altered TSS efficiency effects (see Discussion). When we combined alleles of *sua7-1* or *tfg2Δ146–180* with *ssl2* alleles that shift TSS distributions in opposite directions, interactions were complex but significant epistasis was observed. Consistently, double mutants shifted *ADH1* TSS distributions to similar extent as the *tfg2Δ146–180* single mutant

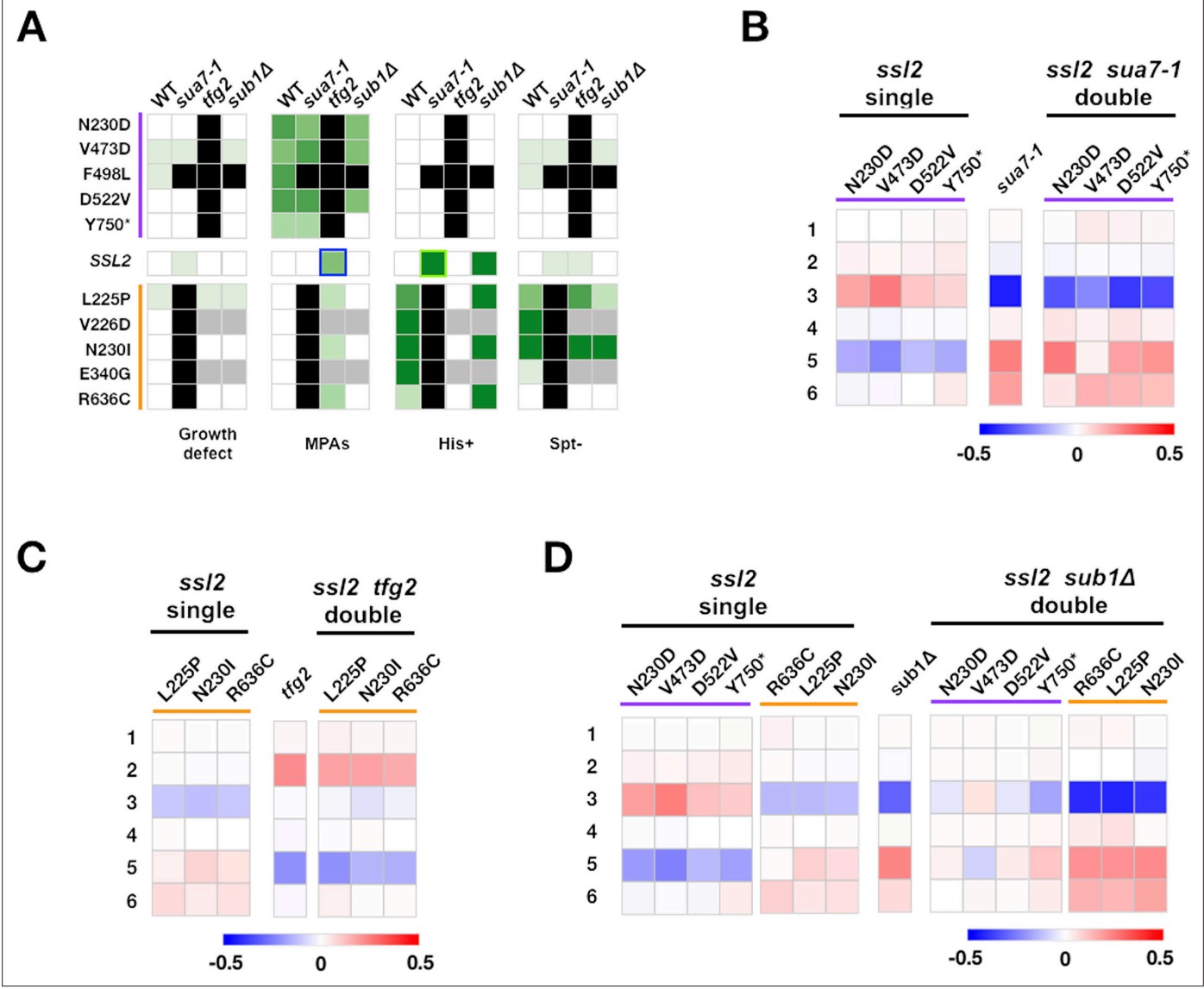

**Figure 7.** Complex genetic interactions between general transcription factor (GTF) initiation alleles suggest multiple distinct activities in initiation by scanning. (**A**) Genetic interactions between *ssl2*, TFIIB, TFIIF, and *sub1* mutants shown as a heatmap indicating phenotypic strength of single and double mutants. Scaling as in *Figure 6F*. (**B–D**) Heatmaps showing quantified *ADH1* primer extension data for *ssl2*, *sua7-1*, *tfg2*, *sub1Δ* single and double mutants. Primer extension as in *Figure 1E*, etc. Mean changes of ≥3 independent biological replicates are shown in the heatmaps.

The online version of this article includes the following figure supplement(s) for figure 7:

**Source data 1.** *Figure 7A* Heatmap data.

**Source data 2.** *Figure 7B* Heatmap data.

**Source data 3.** *Figure 7C* Heatmap data.

**Source data 4.** *Figure 7D* Heatmap data.

**Figure supplement 1.** TFIIB and TFIIF alleles show strong and distinct genetic interaction behavior with *ssl2* alleles.

**Figure supplement 1—source data 1.** *Figure 7—figure supplement 1C* Primer extension gel (annotated).

**Figure supplement 1—source data 2.** *Figure 7—figure supplement 1D* Graph data.

**Figure supplement 1—source data 3.** *Figure 7—figure supplement 1G* Primer extension gel (annotated).

**Figure supplement 1—source data 4.** *Figure 7—figure supplement 1H* Graph data.

**Figure supplement 1—source data 5.** *Figure 7—figure supplement 1C* Primer extension gel (raw).

**Figure supplement 1—source data 6.** *Figure 7—figure supplement 1G* Primer extension gel (raw).

*Figure 7 continued on next page*

*Figure 7 continued*

**Figure supplement 2.** Multiple genetic interactions between *ssl2* and *sub1Δ* alleles.

**Figure supplement 2—source data 1.** *Figure 7—figure supplement 2B* Primer extension gel (annotated).

**Figure supplement 2—source data 2.** *Figure 7—figure supplement 2B* Primer extension gel (annotated).

**Figure supplement 2—source data 3.** *Figure 7—figure supplement 2B* Primer extension gel (annotated).

**Figure supplement 2—source data 4.** *Figure 7—figure supplement 2B* Primer extension gel (annotated).

**Figure supplement 2—source data 5.** *Figure 7—figure supplement 2C* Graph data.

**Figure supplement 2—source data 6.** *Figure 7—figure supplement 2B* Primer extension gel (raw).

**Figure supplement 2—source data 7.** *Figure 7—figure supplement 2B* Primer extension gel (raw).

**Figure supplement 2—source data 8.** *Figure 7—figure supplement 2B* Primer extension gel (raw).

**Figure supplement 2—source data 9.** *Figure 7—figure supplement 2B* Primer extension gel (raw).

(*Figure 7C*, *Figure 7—figure supplement 1*) as predicted for an increase in initiation efficiency buffering against effects of increase in scanning processivity.

Sub1, a conserved factor (yeast homolog of mammalian PC4) was previously found to facilitate Pol II transcription in a variety of ways (*Garavís and Calvo, 2017*; *Calvo, 2018*), to be recruited to the PIC (*Sikorski et al., 2011*), and to alter accessibility of promoter single-stranded DNA, consistent with initiation functions (*Lada et al., 2015*). *sub1Δ* has extensive genetic interactions with initiation factors and itself causes TSSs to shift downstream (*Wu et al., 1999*; *Knaus et al., 1996*; *Braberg et al., 2013*; *Koyama et al., 2008*), though its actual role in initiation is unknown. We previously found *sub1Δ* to confer a His⁺ phenotype for the *imd2Δ::HIS3* initiation reporter (*Malik et al., 2017*) and furthermore found that Pol II GOF alleles appeared epistatic to *sub1Δ*, leading to the proposal that *sub1Δ* effects in initiation were distinct from TFIIB or TFIIF alleles (*Jin and Kaplan, 2014*). Because we have observed similar epistatic interactions between *ssl2* and Pol II alleles, we considered that Sub1 might also be behaving as a scanning processivity factor. Therefore, we predicted the possibility of additive effects between two types of processivity alleles, namely *ssl2* upstream and downstream shifting alleles and *sub1Δ*, if they are acting independently. First, no strong genetic interactions (lethality) were observed between *ssl2* and *sub1Δ* alleles, save for one specific case (*Figure 7A*, *Figure 7—figure*

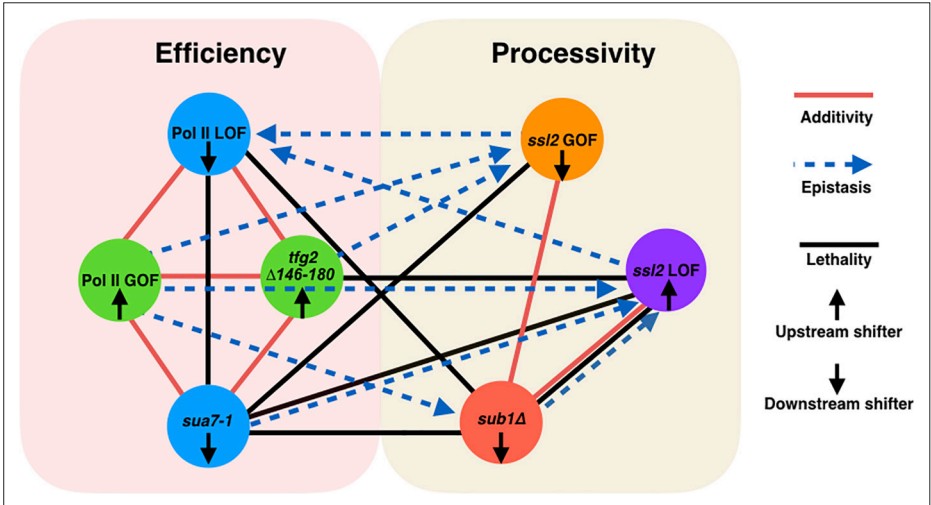

**Figure 8.** Two major functional networks controlling initiation by scanning. In our genetic experiments, additive/suppressive effects are mainly observed between alleles predicted to function by alteration to initiation efficiency (*rpb1* and tested alleles of TFIIB/TFIIF). Multiple lines between classes indicate allele-specific interactions between a factor and individuals of an allele class, for example, *sub1Δ* and *ssl2* LOF alleles. In contrast to interactions within the 'efficiency' network, widespread epistasis was observed between *ssl2* and other factors as predicted for interactions between processivity and efficiency alleles. *sub1Δ* generally shows additivity/suppression with *ssl2* alleles, consistent with it functioning as a scanning processivity factor. Unique lethal interactions between *ssl2* loss-of-function (LOF) upstream shifter F498L and *sua7-1* and *sub1Δ* indicate distinct behavior within the *ssl2* LOF class.

*Zhao et al. eLife 2021;10:e71013. DOI: https://doi.org/10.7554/eLife.71013*

*supplement 2A*). Second, the majority of *sub1Δ* interactions with *ssl2* alleles appear to be additive when examining TSS distributions at *ADH1* as predicted for factors are acting on processivity independently. Third, and notably, we identified allele-specific interactions between *sub1Δ* and specific *ssl2* alleles within classes of *ssl2* allele that until these experiments have not been distinguishable. For example, most upstream shifting *ssl2* alleles were additive with *sub1Δ* for TSS distributions at *ADH1*, resulting in mutual suppression of TSS distribution shifts (*Figure 7D*). In contrast, *sub1Δ* was epistatic to *ssl2* Y750*, suggesting that a putative block to processivity due to C-terminal truncation of Ssl2 can be relieved by *sub1Δ*, and potentially may be due to altered Sub1 function in *ssl2* Y750*. Finally, allele specificity of *ssl2* F498L was revealed by these genetic experiments. This TSS upstream shifting *ssl2* allele was unexpectedly synthetic lethal with both *sua7-1* and *sub1Δ* suggesting heretofore undetected phenotypic differences from other alleles of the same class.

### Two networks controlling initiation by promoter scanning

Results of genetic interaction studies are consistent with two distinct networks controlling TSS selection by scanning (*Figure 8*). Additive/suppressive interactions were observed within networks while specific classes of mutants showed epistatic interactions between networks. One network impinges on Pol II catalysis and initiation efficiency, and genetic analyses suggest that the Pol II active site collaborates with activities of TFIIB and TFIIH in this process, consistent with experiments indicating effects of TFIIB and TFIIF on Pol II catalytic activity (e.g., *Khaperskyy et al., 2008*; *Cabart et al., 2014*; *Sainsbury et al., 2013*). The other, we propose, impinges on scanning processivity through TFIIH with the participation of Sub1. Our genetic interactions also uncover functional connections between TFIIB and TFIIF and Ssl2 that are distinct from Pol II active site mutants. These results support predictions of altered PIC function for TFIIB and TFIIF mutants beyond phosphodiester bond formation and will be interesting to test in biophysical experiments. Extensive epistasis observed between networks (*Figure 8*) supports predictions for how efficiency and processivity should interact during initiation by promoter scanning (*Figure 9*, see Discussion).

### *ssl2* alleles shift positioning of PIC-components genome-wide

We found previously that polar shifts of TSS distribution in Pol II catalytic activity mutants were accompanied with alteration in PIC localization as detected by ChIP-exo (*Qiu et al., 2020*). Shift in PIC components upon alteration to Pol II catalytic activity suggested that extent of scanning might be coupled to Pol II initiation, or that alteration to Pol II initiation kinetics affects observed distributions of GTFs. We performed these same ChIP-exo experiments on Sua7 and Ssl2 for two *ssl2* alleles, N230D and N230I (*Figure 9—figure supplement 1A, B*). Both shifted PIC localization genome-wide with the same polarity as they shift TSS distributions. We note that TAP-tagging Ssl2 confers slight phenotypes on its own and slight enhancement of *ssl2* N230D and slight suppression of *ssl2* N230I (*Figure 9—figure supplement 2*). However, each tagged mutant was compared to the tagged WT and the results are robust and distinct for each mutant. The extent of ChIP-exo shifts were as strong or stronger than Pol II mutant shifts although Pol II mutants have stronger effects on TSS distributions (*Qiu et al., 2020*). These results could be consistent with scanning by TFIIH on DNA uncoupled from the Pol II initiation decision, that is, *ssl2* mutants extend PIC scanning to downstream positions even though initiation has occurred. Such a result would be consistent with the similar behavior for DNA compaction in optical tweezer analysis of initiation wherein dATP-supported reactions (presumptive TFIIH scanning-driven DNA translocation through Ssl2 use of dATP) and NTP-supported initiation reactions (TFIIH translocation and Pol II initiation allowed) have similar behavior (*Fazal et al., 2015*). The extent or mechanism of uncoupling between promoter scanning upon productive initiation is unknown and represents an open question in initiation mechanisms.

## Discussion

Our studies now reveal the impact of altered Ssl2 function on initiation by promoter scanning in *S. cerevisiae*. We find distinct classes of Ssl2 allele that alter initiation genome-wide with distinct behaviors, with many alleles being in highly conserved residues. Our genomic and genetic data support a model wherein Ssl2 function as a DNA translocase can be genetically modulated and this modulation is consistent with TFIIH having either increased or decreased processivity during promoter scanning.

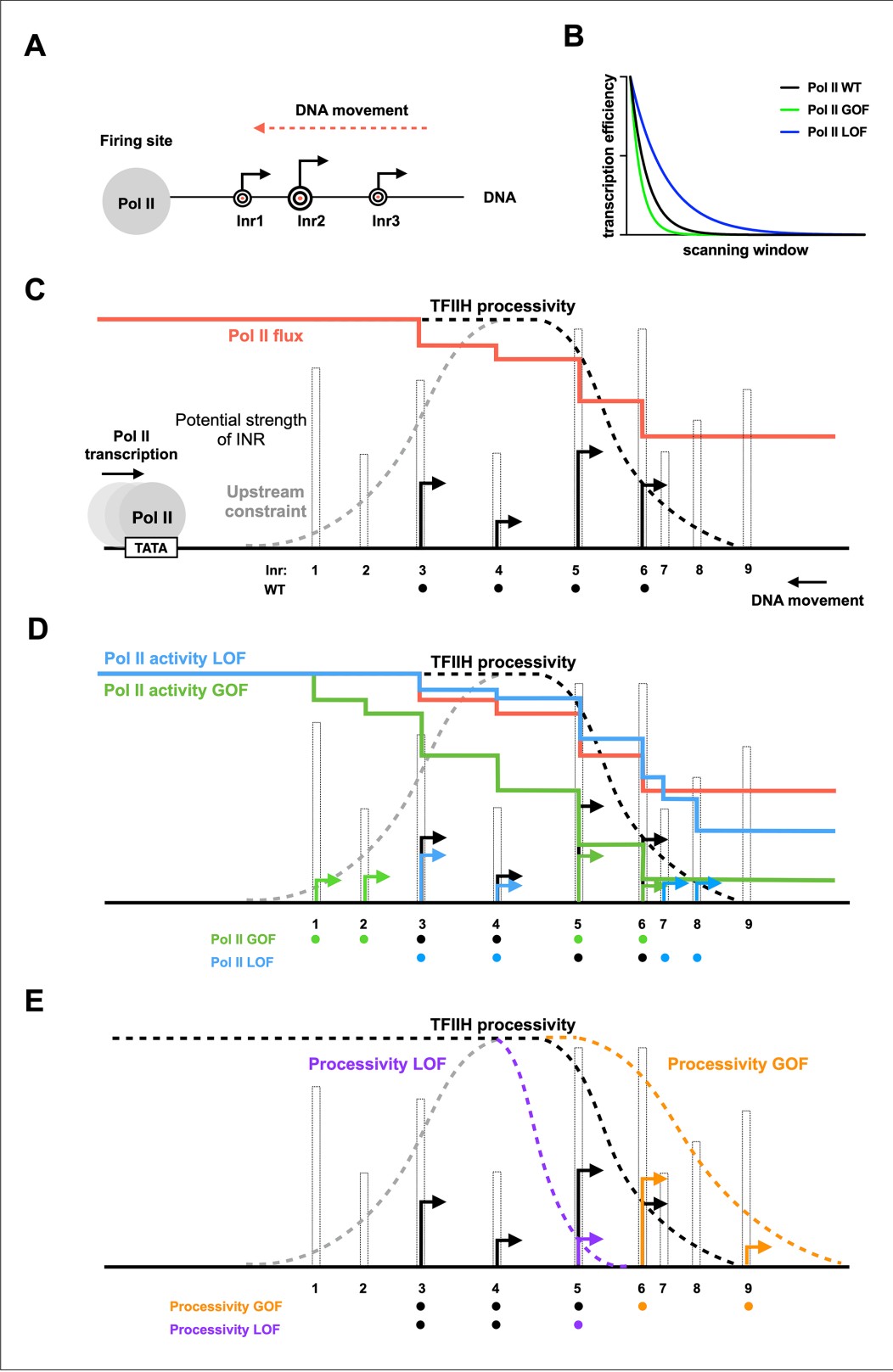

**Figure 9.** Model for interaction between initiation efficiency and scanning processivity. (**A**) The 'Shooting Gallery' model. The polymerase II (Pol II) active site controls initiation efficiency, that is, 'the rate of firing'. TFIIH controls the rate and extent of scanning, that is, 'the speed of target passage and number of targets reached'. (**B**) Reduction in relative transcription start site (TSS) usage as scanning Pol II initiates. As Pol II (wild-type [WT]) scans from upstream

*Figure 9 continued on next page*

*Figure 9 continued*

to downstream, successful initiation at upstream positions will reduce the amount of Pol II continuing to scan downstream. Increasing initiation efficiency at each position as is predicted for increased Pol II catalytic activity will result in a more rapid decrease in observed initiation from upstream to downstream. Conversely, reducing initiation efficiency at each position will flatten observed TSS distribution because more Pol II will reach downstream positions. (**C**) TSS distributions during promoter scanning in the 'Shooting Gallery' model. The TSS distribution (black arrows) of a promoter window can be affected by Pol II catalytic activity, preinitiation complex (PIC) scanning rate and processivity, TSS strength, Pol II flux, and additional observed (upstream limitation on initiation too close to PIC assembly) or potential (downstream limitation through chromatin structure) constraints. (**D**) Effects of Pol II catalytic activity on TSS distributions. Increased Pol II catalytic activity increases the efficiency of upstream TSSs that are encountered by Pol II and decreases the usage of downstream TSSs due to quickly reduced Pol II flux (changes indicated as green arrows). Decreased Pol II catalytic activity decreases TSS efficiency of upstream TSSs encountered by Pol II and increases apparent TSS usage at downstream sites due to failed upstream initiation, resulting in a downstream shifted TSS distribution within a window determined by PIC scanning potential (changes shown as blue arrows). (**E**) Effects of altered scanning processivity on TSS distributions. Increased processivity alleles are hypothesized to increase the probability of Pol II scanning further downstream if Pol II flux remains, thus expanding the scanning window and allowing Pol II usage of downstream TSSs if Pol II flux is not limiting (orange TSS). In contrast, decreased processivity will limit Pol II scanning downstream, truncating the distribution of observed TSSs (purple TSS).

The online version of this article includes the following figure supplement(s) for figure 9:

**Figure supplement 1.** *ssl2* alleles shift preinitiation complex (PIC)-component positioning genome-wide as predicted for mutants altering scanning processivity.

**Figure supplement 1—source data 1.** *Figure 9—figure supplement 1A* Graph data.

**Figure supplement 1—source data 2.** *Figure 9—figure supplement 1B* Graph data.

**Figure supplement 2.** Effects of TAP-tagging *SSL2*.

---

The positioning and genetic behaviors of our allele classes are consistent with one class behaving biochemically as an LOF and therefore truncating the scanning process prematurely and narrowing TSS distributions genome-wide. This is exactly the predicted outcome for a translocase with decreased processivity. Conversely, our other class increases downstream TSS usage and alters PIC localization at promoters by extending it downstream. These behaviors are consistent with increased translocase processivity. Genetic interactions between *ssl2* alleles especially between *ssl2* and GTF mutants and *sub1Δ* suggest further distinctions between allele classes or within allele classes, generating testable predictions for biochemical studies. Putative increased *SSL2* activity alleles are also dominant or codominant genetically, consistent with being able to function on promoters in an increased capacity of some sort.

## Interpretation of Pol II and Ssl2 functions in the Shooting Gallery model for initiation by scanning

We have previously described how Pol II determines the *efficiency* of a TSS in a 'Shooting Gallery' model, where the rate at which a TSS (conceived of as a target) passes the active site, the rate of firing (catalytic activity), and the size of the target (innate sequence strength) together contribute to the probability a target is hit (initiation happens) (*Figure 9A*; *Qiu et al., 2020*; *Kaplan, 2013*). Alteration of enzymatic activities supporting initiation, either the Pol II active site or TFIIH translocation, will have predictable effects on individual TSS usage and the overall TSS distributions when initiation proceeds by scanning. In Pol II mutants with altered catalytic activity that is known to affect transcription efficiency, we observed polar changes to TSS distributions (*Qiu et al., 2020*). Distributions will also necessarily be shaped by an additional factor: Pol II flux. Pol II flux describes the relative number of polymerases encountering a given start site, which has a higher value at upstream TSSs and a lower value at downstream TSSs, resulting in reduced apparent usage at downstream position distinct from their inherent efficiencies (*Figure 9B*). Additionally, the potential upstream and downstream constraints for defining the scanning window will have effects on TSS distributions. Studies suggest that very upstream TSSs close to the presumed location of PIC assembly show reduced transcription initiation (*Faitar et al., 2001*). The physical basis for defining the upstream boundary of the scanning window has not yet been determined. An obvious constraint is the minimum space required for PIC

assembly. Moreover, we hypothesize that downstream constraints for defining the scanning window could be TFIIH's processivity, the +1 nucleosome, or both. Previous single molecule studies suggested that TFIIH drives downstream scanning distances similar in length to the distribution of TSSs at yeast promoters (*Fazal et al., 2015*). We propose that TSS distribution of a promoter is established by the cooperation of Pol II's catalytic activity and TFIIH's processivity for reaching and activating TSSs at promoter sites.

When Pol II has increased catalytic activity, for example, in Pol II catalytic activity GOF alleles, upstream TSSs will increase in efficiency (*Figure 9C and D*, Pol II GOF). In this allele class, usage of downstream TSSs also will decrease due to reduction in Pol II flux reaching downstream sites due to prior initiation. Conversely, when Pol II has decreased catalytic activity, TSSs at upstream sites will be less efficiently used, more slowly reducing Pol II flux (*Figure 9C and D*, Pol II LOF). Inability to initiate earlier in scanning will result in increased TSS usage at downstream sites and a flattening and spreading of the TSS distribution (as demonstrated by an efficiency curve with decreased slope). We hypothesize that alleles with increased processivity (*processivity* GOF allele) will expand the scanning window by allowing the PIC to scan further downstream while attempting initiation, increasing the probability that downstream TSSs are reached during any individual scanning event (*Figure 9E*, processivity GOF). As a consequence, *processivity* GOF alleles increase the potential for scanning downstream but only if Pol II flux (Pol II molecules still scanning) persists to reach those sites. In contrast, a *processivity* LOF allele would limit the Pol II machinery's access to downstream TSSs sites by reducing the scanning window (*Figure 9E*, processivity LOF). Consequently, there would be an upstream shift in TSS distribution compared to WT, without the activation of additional upstream TSSs.

Prior biochemical and more recent structural analyses indicate that TFIIH is a fascinating complex with numerous contacts suggested or predicted to modulate or control TFIIH enzymatic subunits' activities (*Nogales and Greber, 2019*). For example, Ssl2/XPB must be activated during transcription initiation to allow promoter opening. Recent structures suggest that interactions with both Mediator and TFIID may position parts of TFIIH for different functions in initiation (*Abdella et al., 2021*; *Chen et al., 2021*; *Rengachari et al., 2021*). Genetic and biochemical studies also suggest that TFIIH may itself impose a block to initiation that is then relieved by TFIIH activity through Ssl2/XPB (*Alekseev et al., 2017*; *Lin et al., 2005*). Both TFIIH ATPase subunits, Rad3/XPD and Ssl2/XPB, must also be regulated for TFIIH's function in NER with Rad3/XPD held inactive during transcription and released for NER (reviewed in *Greber et al., 2019*; *Nogales and Greber, 2019*). XPB mutations in patients that result in XP are straightforwardly interpreted as conferring NER defects, however transcriptional phenotypes may also be present depending on mutation (*Oh et al., 2006*; *Cleaver et al., 1999*; *Weeda et al., 1997*). Mutations in XPB that cause TTD localize to conserved residues in the XPB N-terminus where we have identified a number of mutations. TTD mutations in XPB have been interpreted as reducing the amount of TFIIH in the cell through potential destabilization, while one appears to impact folding and activity of XPB (*Greber et al., 2019*). Our identification of putative LOF and GOF mutations in this domain in *S. cerevisiae* underscores the idea that observed conservation in this region may control key inputs to Ssl2/XPB activity. The substitutions we have identified are largely in residues conserved from yeast to humans (*Figure 2—figure supplement 3*), and we suggest that these residues detect potential paths for allosteric regulation of Ssl2/XPB. Only a subset of our alleles confer UV sensitivity, suggestive of NER defects (*van Eeuwen et al., 2021*). These are C-terminal and this suggests that our alleles uniquely alter Ssl2 modulation in transcription or that transcriptional functions of Ssl2 are sensitized to defects that do not appreciably lead to UV sensitivity. DNA translocases are the engines for chromatin remodeling and much regulation of chromatin remodelers relates to coupling of ATP hydrolysis to translocation potential (*Clapier et al., 2017*; *Clapier et al., 2016*). Mutations in remodelers that increase or decrease coupling have strong effects on remodeling. We posit that it is likely that a number of our alleles will act through altered coupling of ATPase activity and translocation, with the end result being increase or decrease in scanning processivity. Biochemical studies will reveal specific aspects of TFIIH activity that are altered by these substitutions.

Our putative GOF alleles are concentrated in the TFB2C-like N-terminal domain and DRD of Ssl2. The TFB2C domain has been implicated as a target of Tfb3/Mat1 in restricting Ssl2/XPB activity in holoTFIIH (*Luo et al., 2015*; *Nogales and Greber, 2019*; *Greber et al., 2017*). However, upon assembly into the PIC, Tfb3/Mat1 releases the N-terminus of Ssl2 (*Abdella et al., 2021*; *Schilbach et al., 2017*). This Ssl2 domain is also targeted by Tfb2/p52 (*Schilbach et al., 2021*; *He et al., 2016*;

*Greber et al., 2019*; *Schilbach et al., 2017*), which has long been implicated in modulating Ssl2/XPB activity in addition to assembling it into TFIIH. We have identified one His⁺ allele at the Ssl2-Tfb2 interface, though most are in internal interfaces at the putative nexus of Ssl2 NTD, DRD, and HD1/ATPase lobe 1 (*Figure 2*).

Key open questions relate to how the PIC communicates to Ssl2/XPB to engage and open promoter DNA, what the mechanistic basis for any imposition of initiation block by Ssl2/XBP is, what the basis of its subsequent relief is, and how translocation is terminated upon or subsequent to productive initiation, that is, are the processes coupled in anyway? Aibara, Schilbach et al. have recently imaged the human PIC captured in two states, suggestive of pre- and post-translocation intermediates of TFIIH (*Aibara et al., 2021*). These structures show loss of contact between TFIIH through the MAT1 RING domain and the Pol II stalk/TFIIE in the proposed post-translocated state. This raises an attractive model for uncoupling TFIIH translocase from Pol II after a single translocation step, functionally limiting initiation to the small window of exposed TSSs within reach of the Pol II active site. These structures were from a minimal PIC lacking Mediator and TFIID and therefore it remains to be determined if this translocase is in fact uncoupled, as other potential TFIIH/PIC contacts remain. Other events during initiation may also propagate changes to the PIC, such as lengthening of the nascent RNA to potentially clash with TFIIB and potential subsequent reorganization of the PIC by this event, or due to Pol II CTD phosphorylation. In many organisms, nucleosome-depleted regions promote initiation bidirectionally and these regions are flanked by positioned nucleosomes (*Vo Ngoc et al., 2017*; *Duttke et al., 2015*; *Chen et al., 2016*; *Kaplan, 2016*). Nucleosomes would potentially act as competitors for double-stranded DNA being translocated by TFIIH. Transcription activity drives histone dynamics at promoters, consistent with TFIIH translocation proposed to function akin to a chromatin remodeler (*Tramantano et al., 2016*). How the +1 nucleosome might feed back on scanning or regulate TFIIH translocation is an open question. In the absence of the remodeler RSC's function in yeast, nucleosomes move upstream into normally nucleosome depleted regions and inhibit or narrow TSS usage at a number of promoters (*Klein-Brill et al., 2019*). These results are consistent with nucleosomes competing with TFIIH for promoter DNA. However, these results are conditional on RSC depletion, and it is not clear if promoter nucleosome remodeling under normal conditions obviates the ability for +1 nucleosomes at active promoters to provide a block to scanning or initiation on the edges of their positions.

How Pol II specifies multiple TSSs at individual promoters across eukaryotes has long been an open question. The observation of downstream-located TSSs relative to where DNA melting occurs in *S. cerevisiae* led to the original proposal of a scanning mechanism for TSS identification (*Giardina and Lis, 1993*). In contrast to how Pol II finds TSSs in *S. cerevisiae* for all promoters, in other eukaryotes including other fungi, a scanning process is not required for promoters with defined architecture specified by a TATA element (*Lu and Lin, 2021*; *Breathnach and Chambon, 1981*). These promoters use highly focused TSSs immediately and precisely downstream of the DNA melting sites ~30 bp downstream of the TATA element +1 position. For example, Lu et al. have pinpointed the split between scanning from TATA-promoters and non-scanning species within the Saccharomycetes (*Lu and Lin, 2021*). However, most eukaryotic promoters are TATA-less and use multiple TSSs (*Vo Ngoc et al., 2017*; *Li et al., 2015Kawaji, 2014*; *Saito et al., 2013*; *Nepal et al., 2013*; *Chen et al., 2013*; *Yamashita et al., 2011*; *Hoskins et al., 2011*; *Kawaji et al., 2006*; *Carninci et al., 2005*; *Kadonaga, 2012*; *Haberle and Stark, 2018*). Therefore, whether or not scanning is also a mechanism in higher eukaryotic promoters, or minimally for a subset of eukaryotic promoters or within a specified window, is still an unanswered question as there have been no formal tests of this mechanism. It has been suggested, however, that each individual TSS is recognized as an individual promoter due to sequence signatures apparent in comparison of thousands of TSSs in humans (*Luse et al., 2020*). Very recent results of cryo-EM studies on human PICs, especially in structures visualizing TFIID, indicate that promoter classes may assemble PIC components in distinct fashion within a single organism (*Chen et al., 2021*), yet these assembly pathways similarly position an upstream TSS proximal to the Pol II active site, consistent with proposals that human promoters could contain information for assembling PICs individually (*Luse et al., 2020*). Recent results suggest that there may be plasticity in TSS selection from individual PICs upon mutation of Inr sites in mouse, potentially supporting a type of scanning in mammals (*Chou et al., 2021*). That diverse initiation mechanisms are supported by highly conserved factors suggests that we may yet to find additional unexpected plasticity in initiation across evolution in eukaryotes.

## Materials and methods

### Yeast strains

Yeast strains are derived from a *GAL⁺* of S288C (FY2) (*Winston et al., 1995*). Yeast strains used in this study are listed in *Supplementary file 1*.

### Plasmids and bacterial strains

Bacterial strains and plasmids used in this study are listed in *Supplementary file 2*.

### Yeast media

Yeast media used in this study were made as previously described (*Jin and Kaplan, 2014*; *Malik et al., 2017*; *Kaplan et al., 2012*; *Amberg et al., 2005*). Briefly, YP medium is made of yeast extract (1% w/v; BD) and peptone (2% w/v; BD). Solid YP medium contained bacto agar (2% w/v; BD), adenine (0.15 mM, Sigma-Aldrich), and tryptophan (0.4 mM, Sigma-Aldrich). YPD medium uses YP medium components supplemented with dextrose (2% w/v, VWR). YPRaf medium uses YP medium components supplemented with raffinose (2% w/v, Amresco) and antimycin A (1 mg/ml; Sigma-Aldrich). YPRafGal medium uses YP medium components and supplemented with raffinose (2% w/v), galactose (1% w/v; Amresco), and antimycin A (1 mg/ml; Sigma-Aldrich). Minimal media (SC-) was made with a slightly modified 'Hopkins mix' (0.2 % most amino acids w/v), and supplemented with Yeast Nitrogen Base containing ammonium sulfate (without amino acids, BD), bacto agar (2% w/v; BD), and dextrose (2% w/v, VWR). The original 'Hopkins mix' and the slight modification were as previously described (*Kaplan et al., 2012*; *Amberg et al., 2005*). All amino acids are from Sigma-Aldrich. SC-Leu +5 -FOA (5-fluoroorotic acid) is minimal medium of SC-Leu supplemented with (5-FOA), (1 µg/ml, Gold Biotechnology). SC-Leu+ MPA media is minimal media of SC-Leu supplemented with MPA (20 µg/ml, Sigma-Aldrich, from a 10 mg/ml stock in ethanol). SC-His +3AT is minimal medium of SC-His supplemented with 3-aminotriazole (0.5 mM, Sigma-Aldrich).

### Plate phenotyping and growth heatmaps

Yeast phenotyping assays were performed by spotting 10-fold serial dilutions of saturated YPD-liquid yeast cultures on various solid media, as previously described (*Kaplan et al., 2012*). Yeast cells on various media were cultured at 30° C except for temperature sensitivity phenotypes, which were at 16° C (YPD 16) and at 37° C (YPD 37). Yeast growth on specific media was recorded by taking pictures every 24 hr after an initial 16 hr of growth, from day 2 (40 hr) to day 7 for all media except for YPRaf/Gal (pictures to day 9). Growth phenotypes on specific media were scored on days when WT yeast reached mature colony sizes, as follows: YPD on day 2 (40 hr after spotting); SC-Leu, SC-His, and SC-Trp on day 3 (64 hr); YPRaf on day 4 (88 hr); SC-Lys and SC-Leu+MPA on day 5 (112 hr); and YPRaf/Gal on day 7 (160 hr). To illustrate the strength and distribution of mutants on the two-dimensional structure of Ssl2 (*Figure 2*), growth phenotypes are converted to a numerical score, using the scale 0–6 to indicate the level of growth, where 0 indicates no growth and 6 indicates full growth. The level of growth is positively correlated with the strength of phenotypes for SC-His, SC-Lys, and YPRaf/Gal medium, so the 'growth score' is directly used as 'phenotyping score' for making a heatmap. For other media, the level of growth is negatively correlated with the strength of phenotype, thus growth score 0–6 is inversely converted to the phenotyping strength score 6–0, with 6 growth score converted into phenotyping score 0 to indicate no phenotype, with 5 growth score converted into phenotyping score 1 to show a weak phenotype, and so on. The heatmap uses light to dark color showing weak (phenotyping score 1) to strong phenotypes (phenotyping score 6), no phenotype (phenotyping score 0) has no color.

### Primer extension

To detect putative usage of TSSs in yeast, a primer extension (PE) assay was performed as previously described (*Kaplan et al., 2012*), modifying original protocol in *Ranish and Hahn, 1991*. Briefly, 30 µg of total RNA isolated from yeast cells was used for each PE reaction. A primer complementary to *ADH1* mRNA was end-labeled with gamma-P32 ATP and T4 PNK and annealed to total RNA. Reverse-transcription was then performed by adding M-MLV reverse-transcriptase (Fermentas/ThermoFisher) and RNase Inhibitor (Fermentas/ThermoFisher). RNase A was added to remove RNA after

reverse-transcription. Products were detected by running an 8 % acrylamide gel (19:1 acrylamide:bi-sacrylamide) (Bio-Rad), 1 × TBE, and 7 M urea. PE gels were visualized by phosphorimaging (GE Healthcare or Bio-Rad) and quantified by ImageQuant 5.1 (GE) or Image Lab software (Bio-Rad).

*ssl2* mutant screening *ssl2* mutants were created by PCR-based random mutagenesis coupled with a gap repair. Briefly, mutation of *SSL2* (*ssl2*\*) was accomplished by standard PCR reactions using Taq polymerase (New England Biolabs). *ssl2*\* PCR products were then transformed into yeast along with a linearized pRS315 *SSL2 LEU2* plasmid with most of the WT *SSL2* sequence removed by restriction digest. Leu⁺ transformants were selected. Homologous sequences on each end of the *ssl2*\* PCR products and the gapped *SSL2* vector allowed homologous recombination, resulting in a library of gap-repaired plasmids containing potential *ssl2*\* alleles. Since *SSL2* is essential, these yeast cells are pre-transformed with a pRSII316 *SSL2 URA3* plasmid to support growth, while the genomic *SSL2* was deleted to allow plasmid *SSL2* alleles to exhibit phenotypes. After gap repair, cells retaining pRSII316 *SSL2 URA3* plasmids were killed by replica-plating transformants to medium containing 5-FOA (GoldBio). Yeast cells were then plated on YPD media for growth and replica-plated to a variety of media to screen for mutants that have transcription-related or conditional phenotypes. Plasmids from yeast mutants were recovered and transformed into *Escherichia coli* for amplification, followed by sequencing to identify mutations. Mutant yeast candidates were additionally mated with yeast cells that contain a WT *SSL2 URA3* plasmid to create diploid strains and perform phenotyping again to determine dominance/recessivity of *ssl2* mutations. Plasmid shuffling on diploid strains was performed by adding 5-FOA on the medium so that presumably only presumptive *ssl2*\* was kept. This was followed by an additional phenotyping to determine if the mutant phenotype is plasmid linked or not. All mutants described here were verified by retransformation into a clean genetic background.

## TSS-seq

Yeast cell cultures were grown in triplicates and cells were harvested at mid-log phase at a density of 1 × 10⁷ cells/ml, as determined by cell counting. For *S. cerevisiae* TSS-seq, cells collected from 50 ml of *S. cerevisiae* culture and 5 ml of *Schizosaccharomyces pombe* culture were mixed and total RNA was extracted as described (*Schmitt et al., 1990*). We performed cDNA library construction for TSS-seq essentially as described by *Vvedenskaya et al., 2015*; steps are described as follows; 100 µg of the isolated total RNA was treated with 30 U of DNase I (QIAGEN) and purified using RNeasy Mini Kit (QIAGEN). A Ribo-Zero Gold rRNA Removal Kit (Illumina) was used to deplete rRNAs from 5 µg of DNase-treated RNAs. The rRNA-depleted RNA was purified by ethanol precipitation and resuspended in 10 µl of nuclease-free water. To remove RNA transcripts carrying a 5′ monophosphate moiety (5′-P), 2 µg of rRNA-depleted RNA were treated with 1 U Terminator 5′-Phosphate-Dependent Exonuclease (Epicentre) in the 1 × Buffer A in the presence of 40 U RNaseOUT in a 50 µl reaction at 30° C for 1 hr. Samples were extracted with acid phenol-chloroform pH 4.5 (ThermoFisher Scientific), and RNA was recovered by ethanol precipitation and resuspended in 30 µl of nuclease-free water. Next, to remove 5′-terminal phosphates, RNA was treated with 1.5 U CIP (NEB) in 1 × NEBuffer three in the presence of 40 U RNaseOUT in a 50 µl reaction at 37° C for 30 min. Samples were extracted with acid phenol-chloroform and RNA was recovered by ethanol precipitation and resuspended in 30 µl of nuclease-free water. To convert 5′-capped RNA transcripts to 5′-monophosphate RNAs ligatable to 5′ adaptor, CIP-treated RNAs were mixed with 12.5 U CapClip (Cellscript) and 40 U RNaseOUT in 1 × CapClip reaction buffer in a 40 µl reaction and incubated at 37° C for 1 hr. RNAs were extracted with acid phenol-chloroform, recovered by ethanol precipitation and resuspended in 10 µl of nuclease-free water. To ligate the 5′ adapter, the CapClip-treated RNA products were combined with 1 µM 5′ adapter oligonucleotide s1086 (5′-GUUCAGAGUUCUACAGUCCGACGAUCNNNNNN-3′), 1× T4 RNA ligase buffer, 40 U RNaseOUT, 1 mM ATP, 10 % PEG 8000 and 10 U T4 RNA ligase 1 in a 30 µl reaction. The mixtures were incubated at 16 °C for 16 hr and the reactions were stopped by adding 30 µl of 2 × RNA loading dye. The mixtures were separated by electrophoresis on 10 % 7 M urea slab gels in 1 × TBE buffer and incubated with SYBR Gold nucleic acid gel stain. RNA products migrating above the 5′ adapter oligo were recovered from the gel as described (*Pinto et al., 1994*), purified by ethanol precipitation and resuspended in 10 µl of nuclease-free water. To generate first strand cDNA, 5′-adaptor-ligated products were mixed with 0.3 µl of 100 µM s1082 oligonucleotide (5′-GCCTTGGCACCCGAGAATTCCANNNNNNNNN3′ N = A/T/G/C) containing a randomized 9 nt sequence at the 3′ end, incubated at 65 °C for 5 min, and cooled to 4 °C. A solution containing 4 µl of 5 × First-Strand

buffer, 1 µl (40 U) RNaseOUT, 1 µl of 10 mM dNTP mix, 1 µl of 100 mM DTT, 1 µl (200 U) of SuperScript III Reverse Transcriptase and 1.7 µl of nuclease-free water was added to the mixture. Reactions were incubated at 25 °C for 5 min, 55 °C for 60 min, 70 °C for 15 min, and cooled to 25 °C. 10 U RNase H was added, the mixtures were incubated 20 min at 37 °C and 20 µl of 2 × DNA loading solution (PippinPrep Reagent Kit, Sage Science) were added. Nucleic acids were separated by electrophoresis on 2 % agarose gel (PippinPrep Reagent Kit, external Marker B) to collect species of ~90 to ~ 550 nt. cDNA was recovered by ethanol precipitation and resuspended in 20 µl of nuclease-free water. To amplify cDNA, 9 µl of gel-isolated cDNA was added to the mixture containing 1 × Phusion HF reaction buffer, 0.2 mM dNTPs, 0.25 µM Illumina RP1 primer (5′-AATGATACGGCGACCACCGAGATC TACACGTTCAGAGTTCTACAGTCCGA-3′), 0.25 µM Illumina index primers RPI3-RPI16 (index primers have the same sequences on 5′ and 3′ ends, but different on 6 nt sequence that serves as a barcode (underlined); RPI3: 5′ -CAAGCAGAAGACGGCATACGAGAT<u>GCCTAA</u>GTGACTGGAGTTCCTTGGCA CCCGAGAATTCCA-3′), and 0.02 U/µl Phusion HF polymerase in 30 µl reaction. PCR was performed with an initial denaturation step of 10 s at 98 °C, amplification for 12 cycles (denaturation for 5 s at 98 °C, annealing for 15 s at 62 °C and extension for 15 s at 72 °C), and a final extension for 5 min at 72 °C. Amplified cDNAs were isolated by electrophoresis on 2 % agarose gel (PippinPrep Reagent Kit, external Marker B) and products of ~180 to ~ 550 nt were collected. cDNA was recovered by ethanol precipitation and resuspended in 13 µl of nuclease-free water. Barcoded libraries were pooled and sequenced on an Illumina NextSeq platform in high output mode using custom primer s1115 (5′-CTAC ACGTTCAGAGTTCTACAGTCCGACGATC-3′).

## TSS-seq data processing

Quality control on TSS-seq library FASTQ files was performed to remove reads with low quality using fastq_quality_filter in the FASTX-Toolkit (http://hannonlab.cshl.edu/fastx_toolkit/) package with parameters 'fastq_quality_filter -v -q 20 p 75'. Cutadapt (*Martin, 2011*) was then used to remove the 6 nt 5′ linker with parameter of 'cutadapt -u 6'. The resulting reads were trimmed from 3′ end to 35 nt long with parameter of 'cutadapt -l 35 --minimum-length = 35'. Trimmed reads were mapped to the *S. cerevisiae* R64-1-1 (SacCer3) genome using Bowtie (*Langmead et al., 2009*) with allowance of no more than two mismatches with suppression of non-uniquely mapped reads 'bowtie -p3 -v2 -m1 -q --sam --un', reported in sam files. Uniquely mapped reads were then extracted from sam files using SAMtools (*Li et al., 2009*) and output in bam format 'samtools view -F 4 S -b'. Bam files were then sorted and converted into bed files by SAMtools 'samtools sort -o', and BEDTools (*Schwalb et al., 2011*) 'bedtools bamtobed -cigar'. Customized commands were then used on bed files to identify the genomic coordinate of the 5′ end of each uniquely mapped read 'awk 'BEGIN{FS = OFS = "\t"} $6=="+" {$3=$2 + 1} $6=="-" {$2=$3–1} {print}''. BEDTools was then used to determine pileup (TSS coverage) across the genome with parameters of 'bedtools genomecov -g R64.new. genome -i -bg -strand -' and 'bedtools genomecov -g R64.new.genome -i -bg -strand +', resulting in stranded **bedGraph** files. FASTQ files of individual library were directly processed or contacted by strains/mutants to generate bedGraph files for correlation analysis. For each of 5979 selected yeast promoters, TSS usage was examined within 401 nt wide window, spanning 250 nt upstream and 150 nt downstream of the previously annotated median TSS. Using customized bash and R scripts, TSS coverage from the bedGraph files of a library or a mutant were assigned into the defined windows to generate a 401 × 5979 TSS **count table**, with each row representing one of the 5979 promoters in the same order as the promoter annotation file, each column represents a promoter position, and the number in each cell representing 5′ ends mapping to that position. These count tables were stored in csv files. Using customized R script and the count table of a library or a mutant, an **expression-spread-median file** containing promoter expression, median TSS position of the promoter, TSS spread of the promoter was generated. The median TSS position was defined as the actual TSS containing the 50th percentile of the promoter window. The spread of TSS, which measures the width of the middle 80 % of TSS distribution, was calculated by subtracting positions of 10th percentile and 90th percentile of TSS counts in 401 nt promoter window and adding 1. The positions of 10th percentile and 90th percentile of TSS counts in each promoter window were also stored in this expression-spread-median file. The streamlined codes to generate bedGraph files, the prompter annotation file, and the customized scripts to generate count tables and expression-spread-median files can be found at the GitHub repository https://github.com/Kaplan-Lab-Pitt/Ssl2_scanning.

## TSS correlation

TSS coverage data in library-based bedGraph files were used to examine the correlation between TSS libraries by pairwise comparison. A custom R script was used to filter bedGraph files to examine genome positions with greater than two counts in each library. $Log_2$ transformed TSS counts at the same genomic location in two examined libraries were plotted for all TSS sites to create a heat scatter plot using the LSD R package (*Schwalb et al., 2011*) and the Pearson correlation coefficient was calculated. The correlation coefficients deriving from all pairwise comparisons were plotted by a web-based heatmap tool Morpheus (https://software.broadinstitute.org/morpheus/) and clustered by Euclidean distance. Replicates with correlation coefficient greater than 0.85 and the shortest Euclidean distance to each other in the clustering analysis among all the analyzed libraries were recognized as having good sequencing reproducibility and used for downstream analysis.

## TSS count table and heatmaps

For each of 5979 selected promoters, TSS usage was examined within a 401 nt wide window, spanning 250 nt upstream and 150 nt downstream of the previously annotated median TSS (*Qiu et al., 2020*). Using BEDTools and customized R scripts, TSS coverage from the bedGraph files of a library or mutant were assigned into the defined windows to generate a 401 × 5979 TSS count table, with each row representing one of the 5979 promoters, each column represents a promoter position, and the number in each cell representing 5' ends mapping to that position. The count table was filtered to keep data from n = 4392 promoters with ≥100 sequence reads on average per WT library and used for downstream analyses. The filtered count table was row-normalized to get the relative TSS usage at each promoter position. TSS distribution differences were determined by subtracting normalized WT data (concatenated from both *RPB1* and *SSL2* WT libraries) from normalized mutant data and visualized using heatmaps (Morpheus).

## TSS metrics

Data analyses for TSS distributions were based on customized R scripts and results were plotted in GraphPad Prism 8 (https://www.graphpad.com/scientific-software/prism/) unless otherwise indicated. The median TSS position was defined as the actual TSS containing the 50th percentile of the TSS distribution and was determined for each promoter. 'TSS shift' represents the difference in nucleotide of the median TSS position for each promoter between two libraries or mutants. The distributions of TSS shifts for n = 4392 promoters with ≥100 sequence reads on average per WT library or selected promoter classes in each library or mutant were illustrated by both heatmap (Morpheus) and boxplots.

## TSS spread

The spread of TSS, which measures the width of the middle 80 % of TSS distribution, was calculated by subtracting positions of 10th percentile and 90th percentile of TSS reads in 401 nt promoter window and adding 1. TSS spread of selected promoters are shown in boxplot and compared between libraries by performing one-way ANOVA. The differences of TSS spreads between WT and the mutant for selected promoter classes were presented in heatmap (Morpheus) and/or boxplot.

## ChIP-exo data processing

ChIP-exo data processing was performed as described by Rossi et al. in Nature Communications, 2018, and Qiu et al. in Genome Biology, 2020. Briefly, ChIP-exo libraries were sequenced on a NextSeq 500 in paired-end mode to generate 40 (read1) × 36 bp (read2) reads. Reads passing Q30 quality threshold were then aligned to the sacCer3 genome using the BWA-MEM alignment algorithm (v0.7.9a) with default parameters (*Li, 2013*). After alignment, PCR duplicates were removed using Picard and SAMtools assuming unique combinations of read1 and read2 were PCR duplicates. Using ScriptManager v0.12 (https://github.com/CEGRcode/scriptmanager, RRID:SCR_021797), BAM files of a library were assigned into two **401 × 5979 matrices and saved in CDT files**, which stores counts of 5' position of protein binding on top and bottom strands, respectively. The same as in TSS-seq data analysis, each row of 401 × 5979 matrix representing one of 5979 promoters and each column representing a position in the 401 nt promoter window. These **401 × 5979 matrices were also saved in csv format**. Matrices from the same mutant were combined into a single matrix by adding counts in library matrices at the same dimension and saved in csv files. Similar to TSS-seq analysis, the customized R

script and the matrix of a library or a mutant were used to generate an **expression-spread-median file** containing promoter expression, median 5' position of protein binding, the binding site of the spread of the promoter, and saved in txt files.

## Availability of data and materials

Genomics datasets generated in the current study are available in the NCBI BioProject and SRA, under the accession numbers of PRJNA681384 and SRP295731, respectively. The processed genomic data files are available in GEO, under the accession number of GSE182792. The streamlined commands to generate TSS-seq bedGraph files, count tables, tables of expression, spread, and median TSS can be found at https://github.com/Kaplan-Lab-Pitt/Ssl2_scanning, (copy archived at swh:1:rev:fdcccee50e4b6b801048c163d1ac71585958aec6, *Zhao, 2021*). ChIP-exo and data analysis was performed as described by *Rossi et al., 2018* and *Qiu et al., 2020*. Source data files are listed in *Supplementary file 3*.

## Acknowledgements

The authors thank Chenxi Qiu for R scripts used in statistical analysis of Kruskal-Wallis test with Dunn's correction and Mann-Whitney U test. We are deeply grateful to Scott Kuerten and Fred Hyde (Illumina) for advice and the gift of legacy RiboZero reagents for yeast rRNA removal.

## Additional information

### Competing interests

B Franklin Pugh: BFP has a financial interest in Peconic, LLC, which utilizes the ChIP-exo technology implemented in this study and could potentially benefit from the outcomes of this research.. The other authors declare that no competing interests exist.

### Funding

| Funder | Grant reference number | Author |
| --- | --- | --- |
| National Institute of General Medical Sciences | R01GM120450 | Craig D Kaplan |
| National Institute of General Medical Sciences | R01GM059055 | B Franklin Pugh |
| National Institute of General Medical Sciences | R35GM118059 | Bryce E Nickels |
| National Institute of General Medical Sciences | R01GM097260 | Craig Kaplan |

The funders had no role in study design, data collection and interpretation, or the decision to submit the work for publication.

### Author contributions

Tingting Zhao, Conceptualization, Data curation, Formal analysis, Investigation, Validation, Visualization, Writing – original draft, Writing – review and editing; Irina O Vvedenskaya, Investigation, Writing – review and editing; William KM Lai, Data curation, Investigation; Shrabani Basu, Investigation; B Franklin Pugh, Funding acquisition, Resources; Bryce E Nickels, Funding acquisition, Resources, Supervision; Craig D Kaplan, Conceptualization, Formal analysis, Funding acquisition, Project administration, Resources, Supervision, Visualization, Writing – review and editing

### Author ORCIDs

Shrabani Basu http://orcid.org/0000-0001-5096-5490
B Franklin Pugh http://orcid.org/0000-0001-8341-4476
Bryce E Nickels http://orcid.org/0000-0001-7449-8831
Craig D Kaplan http://orcid.org/0000-0002-7518-695X

Decision letter and Author response
Decision letter https://doi.org/10.7554/eLife.71013.sa1
Author response https://doi.org/10.7554/eLife.71013.sa2

## Additional files

### Supplementary files
• Supplementary file 1. Yeast strains used in this study.
• Supplementary file 2. Bacterial strains and plasmids used in this study.
• Supplementary file 3. List of sources data files and their association with figures in this study.
• Transparent reporting form

### Data availability
Genomics datasets generated in the current study are available in the NCBI BioProject and SRA, under the accession numbers of PRJNA681384 and SRP295731, respectively. The processed genomic data files are available in GEO, under the accession number of GSE182792. The streamlined commands to generate TSS-seq bedGraph files, count tables, tables of expression, spread and median TSS can be found at https://github.com/Kaplan-Lab-Pitt/Ssl2_scanning, (copy archived at https://archive.software-heritage.org/swh:1:rev:fdcccee50e4b6b801048c163d1ac71585958aec6).

The following dataset was generated:

| Author(s) | Year | Dataset title | Dataset URL | Database and Identifier |
|---|---|---|---|---|
| Zhao T, Kaplan CD | 2020 | Function of Ssl2/TFIIH in RNA Polymerase II Transcription Start Site Scanning | https://www.ncbi.nlm.nih.gov/bioproject/?term=PRJNA681384 | NCBI BioProject, PRJNA681384 |
| Zhao T, Kaplan CD | 2021 | Ssl2/TFIIH Function in Transcription Start Site Scanning by RNA Polymerase II in *Saccharomyces cerevisiae* | https://www.ncbi.nlm.nih.gov/geo/query/acc.cgi?acc=GSE182792 | NCBI Gene Expression Omnibus, GSE182792 |
| Zhao T, Kaplan CD | 2020 | Function of Ssl2/TFIIH in RNA Polymerase II Transcription Start Site Scanning | https://www.ncbi.nlm.nih.gov/sra?term=SRP295731 | NCBI Sequence Read Archive, SRP295731 |

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
