## [Decision Letter]

**Acceptance summary:**

Kaplan and colleagues build upon their earlier work using genetic phenotypes to find and analyze mutations that determine how mRNA start sites are chosen. Here they provide convincing genetic evidence supporting the model that TFIIH pushes downstream DNA back into the RNA polymerase active site, creating a window within which the polymerase can choose particular start sites. This will primarily interest those in the transcription field thinking about initiation mechanisms.

**Decision letter after peer review:**

Thank you for submitting your article "Ssl2/TFIIH Function in Transcription Start Site Scanning by RNA Polymerase II in *Saccharomyces cerevisiae*" for consideration by *eLife*. Your article has been reviewed by 3 peer reviewers, one of whom is a member of our Board of Reviewing Editors, and the evaluation has been overseen by Kevin Struhl as the Senior Editor. The reviewers have opted to remain anonymous.

Essential revisions:

1) Provide a more general description of TSS-seq to account for a broad readership and re-analyze data using altered cutoff filters to determine the impact of those filters on the observed variability of transcriptional shifts.

2) Deposit processed data into GEO to enable easier re-use by the scientific community.

3) Improve overall readability of the manuscript. There was a unified sense of too much density by all of the reviewers to the point that impact could be reduced. The authors are encouraged to stream line text and figures throughout the entire paper to make messaging more clear.

*Reviewer #1 (Recommendations for the authors):*

Overall, the strength of this study is readily apparent and that is derived from the level of detail the authors had in their analysis to make it is thorough as possible. I commend the authors on their efforts. However, this same strength is also the primary weakness of the paper. The presentation is so dense that it is an exhausting study to get through. The density is apparent at all levels including the number of figures, figure panels, and the writing in general. The study would have much higher impact if it was more streamlined. While it is not my habit to display editorial activism as a reviewer, I feel this paper would benefit from improved presentation. Along these same lines, as a general comment, the authors could significantly improve the messaging of this paper by including short conclusion/wrap-up sentences for each subsection to inform the reader of the overall importance of the data. As it stands now, most sections abruptly end with the description of the last panel so a reader is left trying to determine the take-home point. I provide a list of additional suggestions for the authors to consider to make the overall narrative more digestible for a general readership.

1) Figure 1: The authors do not introduce what 'lys2-128 Spt-' phenotype actually is. They are reliant on their previous work but there is not enough description as to what the molecular basis of this is so one cannot ascertain its importance. The authors could consider removing it altogether as none of the mutants appeared to display this phenotype.

2) Figure 1: The authors should consider moving panel E to the supplement. I understand its value as a positive control, but the general concept of 'start site shifting' is straightforward and their actual data are clear. By removing it, the individual panels of the figure can be larger.

3) Figure 1: I am having trouble understanding the quantitative basis for the author's conclusion that ssl1 mutants (DEAD and 508) are different than the Pol II mutant (E1103G). Comparing bins 2 and 5, for the E1103G the change relative to wild-type is ~0.25 and ~-0.25 respectively. Whereas the DEAD is ~0.12 and ~-0.15 and 508 is ~0.12 and ~-0.1. These seem really similar. Can the authors provide some statistical evidence that this is the trend or just dampen their claim?

4) Figure 1: I am not sure what the XPB mutants add to the story. The authors should add rationale as to why these mutants are important to analyze and how the results fit into the greater narrative.

5) Figure 2: the authors introduce Spt- phenotypes as 'transcriptional related' phenotype but this is not clear what this phenotype is. It appears to be related to His+, but the authors introduce it without a reasonable explanation. This is especially critical because the authors are contrasting the lack of overlap between Spt- and MPAs phenotypes in the ssl mutants.

6) Figure 3A is a well-constructed schematic but it would be helpful to match the color-coding in panels 3C/D in order to understand three-dimensionally where the mutant phenotypes reside. The authors do point some of this out in the text but it is hard to ascertain this looking at the figure.

7) Figure 4: the choice of coloring could be improved. It is very difficult to see differences, particularly in Figure 4B.

8) I would recommend either shifting the positions of Figures 8 and 9 or moving 9 to the supplement. I also feel that the model presented in Figure 8 is difficult to follow as there is only limited reference to it in the results. The authors could consider featuring a model as the focus of their opening paragraph of the discussion. This would nicely tie together all aspects of the study.

*Reviewer #2 (Recommendations for the authors):*

1. I would recommend expanding the description of the TSS-Seq analysis. In particular, I would recommend expanding on the filters that the authors might have applied to select the gene TSS to investigate. Depending on the sequencing depth and read threshold used, the variability for the shift or with of the investigated TSS could change dramatically. I do not think that will alter dramatically their results but will enable its reanalysis.

2. I would also encourage to add the numeric data associated to the different heatmap show in the manuscript (i.e., Figure 4 and 5). That will facilitate the reuse of the authors data by the scientific community.

3. In the same line, I would encourage the authors to deposit in GEO the processed TSS files (e.g., BedGraph, BigWig or similar). Currently in SRA only the raw data are available, and significant work is needed to make them reusable. Providing browsable filed with IGV as is standard for all GEO deposits will further increase the impact of the author's work.

---

## [Author Response]

Essential revisions:1) Provide a more general description of TSS-seq to account for a broad readership and re-analyze data using altered cutoff filters to determine the impact of those filters on the observed variability of transcriptional shifts.

Thanks for this comment. We have clarified our methodology and approach. Our edited data processing methods are at the end of this response for reference. We have provided a few figures here in this rebuttal to address concerns that shifts potentially could be artifacts of reduced sampling (in the case of our data, they are not) and note how presented data argue against shifts being artifact of undersampling. First, there is no systematic undersampling of any individual mutant relative to WT and the threshold used is on the quite conservative side. Our current threshold was promoters with ≥100 reads on average per library in WT. Given that there are not strong effects in relative read numbers per promoter, this results in, for aggregated libraries per genotype, ~200 reads/promoter window and for many strains, >300 reads per promoter window. Second, heat maps in Figure 4 and Figure 4 —figure supplement 1 show effects across all expression levels as the y axes within promoter groups are rank ordered by WT read counts. Note that polar effects on TSS distribution are observed regardless of expression level (we show average read levels for promoters by expression level in deciles for WT and two representative mutants in Author response image 1, while Author response image 2 shows that average shifts/promoter by expression bins in box plots are insensitive to expression level). Figure 4 —figure supplement 2 shows that shifts are reproducible at the level of individual libraries, both by heat map and PCA analysis. Finally, effects seen at ADH1 (very highly expressed) by primer extension for all genotypes are recapitulated by TSS-seq across all expression levels.

**Author response image 1. sa2fig1:** Example of average reads per promoter per library for WT and two representative ssl2 alleles. The dotted line indicates that just over 70% of promoters are included in our “Average of 100 reads per WT library” cutoff. Mutant libraries are not undersampled by this cutoff as coverage was similar across libraries and there were not extensive differences in read fractions/promoter (mRNA 5’ end levels). Box plots are Tukey plots.

**Author response image 2. sa2fig2:** TSS-shift in ssl2 alleles relative to WT is apparent across all expression deciles. Our current cutoff is just within Decile 8. There would be essentially no effect on results if our cutoff only included top 20% of expressed genes. Box plots are Tukey plots.

2) Deposit processed data into GEO to enable easier re-use by the scientific community.

We apologize that this was not done earlier and it was in process while our manuscript was being reviewed. We've deposited our processed data to GEO under the accession number GSE182792. For TSS-seq data, we've deposited bedgraph files for each individual sequencing library or compiled libraries for each mutant that store TSS counts at each genomic location. We've also deposited 401×5979 TSS count tables that store TSS counts in 401-nt promoter window of 5979 selected promoters. We additionally deposited “expression-spread-median” files for each library or mutant that stores read counts (expression) of 5979 promoters; spread, which measures the width of the middle 80% of TSS distribution; and median TSS position of each 401-nt promoter window. For ChIP-exo data, we've deposited 401×5979 matrices and as both CDT and CSV files, which store counts of 5’ position of sequence reads (the positions representing blocks caused by protein-DNA crosslinks). Our edited data processing methods are at the end of this response for reference.

3) Improve overall readability of the manuscript. There was a unified sense of too much density by all of the reviewers to the point that impact could be reduced. The authors are encouraged to stream line text and figures throughout the entire paper to make messaging more clear.

We have striven to streamline the text while also adding key conclusion statements to paragraphs to allow key points to be more strongly emphasized. We have rewritten the genetics section to the extent we feel treats the work succinctly and focuses more on summarization while keeping the data in the manuscript. In addition, we have attempted to address all specific points of each reviewer as noted below.

Reviewer #1 (Recommendations for the authors):Overall, the strength of this study is readily apparent and that is derived from the level of detail the authors had in their analysis to make it is thorough as possible. I commend the authors on their efforts. However, this same strength is also the primary weakness of the paper. The presentation is so dense that it is an exhausting study to get through. The density is apparent at all levels including the number of figures, figure panels, and the writing in general. The study would have much higher impact if it was more streamlined. While it is not my habit to display editorial activism as a reviewer, I feel this paper would benefit from improved presentation. Along these same lines, as a general comment, the authors could significantly improve the messaging of this paper by including short conclusion/wrap-up sentences for each subsection to inform the reader of the overall importance of the data. As it stands now, most sections abruptly end with the description of the last panel so a reader is left trying to determine the take-home point. I provide a list of additional suggestions for the authors to consider to make the overall narrative more digestible for a general readership.1) Figure 1: The authors do not introduce what 'lys2-128 Spt-' phenotype actually is. They are reliant on their previous work but there is not enough description as to what the molecular basis of this is so one cannot ascertain its importance. The authors could consider removing it altogether as none of the mutants appeared to display this phenotype.

We have adjusted the text to make this more clear. In fact, a specific subset of *ssl2* alleles do show this phenotype. This was especially surprising because this same phenotype is found for Pol II alleles, but with Pol II alleles that shift TSSs in the opposite direction. This was our initial observation that suggested that Pol II and *ssl2* alleles may function differently. We have more clearly indicated this, e.g:

“The Spt^-^ phenotype reporter used in our strains, *lys2-128∂*, detects activation of a TSS within a Ty1 ∂ element at the 5’ end of the *LYS2* gene^59^. […] However, in our identified *ssl2* alleles, none of the Spt^-^ mutants also conferred MPA^S^ (Figure 2A). These observations together are consistent with distinct effects on structure and function in *ssl2* mutant classes.”

2) Figure 1: The authors should consider moving panel E to the supplement. I understand its value as a positive control, but the general concept of 'start site shifting' is straightforward and their actual data are clear. By removing it, the individual panels of the figure can be larger.

We have revised this figure as suggested.

3) Figure 1: I am having trouble understanding the quantitative basis for the author's conclusion that ssl1 mutants (DEAD and 508) are different than the Pol II mutant (E1103G). Comparing bins 2 and 5, for the E1103G the change relative to wild-type is ~0.25 and ~-0.25 respectively. Whereas the DEAD is ~0.12 and ~-0.15 and 508 is ~0.12 and ~-0.1. These seem really similar. Can the authors provide some statistical evidence that this is the trend or just dampen their claim?

Sorry for the confusion here. In the primer extension results, there is a very large effect where E1103G appears to activate TSSs in bin 2, thus it shows the largest relative increase in bin 2 TSSs, which are upstream from the most used ADH1 TSSs (bin 3 and bin 5). For *ssl2* mutants, you see the largest change is a shift in distribution between the normally used TSSs (bin 3 and bin 5), i.e. the increase is in bin 3 not bin 2. We have clarified this in the text.

4) Figure 1: I am not sure what the XPB mutants add to the story. The authors should add rationale as to why these mutants are important to analyze and how the results fit into the greater narrative.

We have added rationale along the lines of the following:

“We additionally constructed and tested human disease-related XPB mutations^51-54^ in the yeast *SSL2* system, together with mutations in the ultra-conserved Arginine-Glutamic Acid-Aspartic Acid (RED) motif. […] The lethal phenotypes of RED motif substitutions in *ssl2* revealed their essential roles in *S. cerevisiae*. These results suggest that a subset of human disease alleles can alter TFIIH functions when placed in the yeast system.”

5) Figure 2: the authors introduce Spt- phenotypes as 'transcriptional related' phenotype but this is not clear what this phenotype is. It appears to be related to His+, but the authors introduce it without a reasonable explanation. This is especially critical because the authors are contrasting the lack of overlap between Spt- and MPAs phenotypes in the ssl mutants.

Please see response to Reviewer 1 point 1.

6) Figure 3A is a well-constructed schematic but it would be helpful to match the color-coding in panels 3C/D in order to understand three-dimensionally where the mutant phenotypes reside. The authors do point some of this out in the text but it is hard to ascertain this looking at the figure.

We take this comment to refer to Figure 2 and not Figure 3. We previously had some difficulty in this figure because we were modeling the Ssl2 NTD based on the XPB NTD, and this complicated each panel. We have updated Figures 2 A,C,D using the just published Cramer high resolution PIC and no longer requiring an overlay of XPB and Ssl2 structures, the figure should be more clear. We have also color-coded allele annotations based on Ssl2 domain (DRD, HD, NTD) to make their locations more apparent.

7) Figure 4: the choice of coloring could be improved. It is very difficult to see differences, particularly in Figure 4B.

This was due likely to averaging as there were more points in the heat map than resolution of the figure. We have opted to overcome this by a more sensitive color scaling but have noted in figure legends that color scale was reduced for the purposes of making the global trend more visually obvious.

8) I would recommend either shifting the positions of Figures 8 and 9 or moving 9 to the supplement. I also feel that the model presented in Figure 8 is difficult to follow as there is only limited reference to it in the results. The authors could consider featuring a model as the focus of their opening paragraph of the discussion. This would nicely tie together all aspects of the study.

We have moved figure 9 to the supplement and have tried to emphasize more strongly the use and discussion of the model (new figure 9) in the discussion while making more clear that Figure 8 summarizes a large amount of interaction data, which we attempt to do a better job of discussing.

Reviewer #2 (Recommendations for the authors):1. I would recommend expanding the description of the TSS-Seq analysis. In particular, I would recommend expanding on the filters that the authors might have applied to select the gene TSS to investigate. Depending on the sequencing depth and read threshold used, the variability for the shift or with of the investigated TSS could change dramatically. I do not think that will alter dramatically their results but will enable its reanalysis.

We apologize for not being as explicit as needed on the details of the methodology. We have addressed this point as part of response to essential revisions. Absolutely the reviewer is astute in that the read depth/threshold are quite conservative and undersampling bias if present (it is not) could add noise to TSS shifts, but not likely bias in one direction or another, while if thresholds were indeed too low, TSS spreads could be affected by systematic undersampling in one strain vs another. The effects observed are present across all expression levels and no individual mutant is undersampled relative to WT.

2. I would also encourage to add the numeric data associated to the different heatmap show in the manuscript (i.e., Figure 4 and 5). That will facilitate the reuse of the authors data by the scientific community.

We absolutely agree and this are included with the resubmitted manuscript. These were in process while manuscript was under review (see description above- heat map data are provided as source data files along with all processed genome data in a GEO submission).

3. In the same line, I would encourage the authors to deposit in GEO the processed TSS files (e.g., BedGraph, BigWig or similar). Currently in SRA only the raw data are available, and significant work is needed to make them reusable. Providing browsable filed with IGV as is standard for all GEO deposits will further increase the impact of the author's work.

This was also in process while manuscript was under review. We have established a GEO deposit under accession number GSE182792.